mathematical modelling/computational biology/behaviour

acoustic communication, nonlinear dynamics, selective attention, Bayesian approach

**Author for correspondence:**
Ikkyu Aihara
e-mail: aihara@cs.tsukuba.ac.jp

# Interaction mechanisms quantified from dynamical features of frog choruses

Kaiichiro Ota[1,2], Ikkyu Aihara[3] and Toshio Aoyagi[2,4]

[1]Cybozu, Inc., Tokyo, Japan
[2]JST CREST, Tokyo, Japan
[3]Graduate School of Systems and Information Engineering, University of Tsukuba, Tsukuba, Japan
[4]Graduate School of Informatics, Kyoto University, Kyoto, Japan

 IA, 0000-0002-2111-3050

We employ a mathematical model (a phase oscillator model) to describe the deterministic and stochastic features of frog choruses in which male frogs attempt to avoid call overlaps. The mathematical model with a general interaction term is identified using a Bayesian approach, and it qualitatively reproduces the stationary and dynamical features of the empirical data. In addition, we quantify the magnitude of attention paid among the male frogs from the identified model, and then analyse the relationship between attention and behavioural parameters using a statistical approach. Our analysis demonstrates a negative correlation between attention and inter-frog distance, and also suggests a behavioural strategy in which male frogs selectively attend to a less attractive male frog (i.e. a male producing calls at longer intervals) in order to more effectively advertise their superior relative attractiveness to females.

## 1. Introduction

Animals show various types of behaviour in the form of aggregations. For instance, fish and birds construct a robust flexible school or flock [1]. To maintain the group, members need to synchronize their velocity and direction. On the other hand, various animals (e.g. mammals, birds, anurans and insects) aggregate in a certain area and use acoustic signals for communication, mate identification and attraction [2–5]. Experimental studies demonstrate that these animals tend to alternate their acoustic signals [4–7]. Because such alternating behaviour reduces the acoustic interference of their signals, it may facilitate effective communication and assessment of signallers within the aggregation. Thus, synchronization and alternation are common in the aggregations of animals, and can indicate the quality of their behaviour.

**Figure 1.** Audio data on the choruses of three male Japanese tree frogs. (*a*) Photograph of a calling frog. (*b*) Tri-phase synchronization of three frogs. They successfully avoid call overlaps. (*c*) Clustered anti-phase synchronization of three frogs. Each pair of nearest neighbours avoid call overlaps while a distant pair (i.e. the pair of frogs 1 and 3) overlap their calls. These figures are obtained from the empirical data of our previous study [19].

To synchronize or alternate behaviour, animals must recognize a specific target in an aggregation. Such selective attention is reported in various systems. For example, humans pay attention to one of the talking people in noisy environment like a party [8]; fish and birds attend their neighbour when forming a school or flock [9,10]; bats pay attention to specific targets during prey capture [11]; male frogs pay attention to specific sound sources when advertising themselves by calling [4,5,12–15]. To understand the roles of selective attention, it is essential for us to quantify interaction mechanisms and evaluate these with respect to the behaviour of interacting animals.

This study aims to quantify interaction mechanisms in choruses of male Japanese tree frogs (*Hyla japonica*) by estimating the parameters of a mathematical model (a phase oscillator model [16]) from empirical data. Japanese tree frogs are observed widely in Japan, and often breed in paddy fields from April to July [17]. The male frogs form a lek, and produce calls to attract conspecific females. Laboratory experiments with multiple male Japanese tree frogs demonstrated that they avoid overlapping calls through anti-phase synchronization of two frogs, tri-phase synchronization of three frogs, and clustered anti-phase synchronization of three frogs [18,19] (see figure 1 for examples of three frogs). Moreover, field observations revealed two-cluster synchronization with a larger number of the male frogs in which each pair of nearest neighbours tend to call alternately in their natural habitat [20,21]. Because the temporal overlap of acoustic signals can mask or degrade the information included in each call [22], these alternating behaviours [18–21,23] can help male frogs more effectively advertise themselves to conspecific females (*H. microcephala* and *H. versicolor* [24]; *Engystomops pustulosus* [25]).

This paper is organized as follows: (i) a mathematical model (a phase oscillator model [16]) is identified for each pair of male Japanese tree frogs based on our empirical data using a Bayesian approach, (ii) the attention paid among the male frogs is quantified on the basis of the identified model, and (iii) the relationship between the quantified attention and behavioural parameters is statistically analysed. Note that the phase oscillator model is theoretically derived from general classes of periodic oscillators [16], and can describe synchronized features in coupled biological oscillator including calling behaviour of male frogs [18–20,26]. Thus, our main goal in this study is to identify the interaction mechanisms of frog choruses within a reliable mathematical framework, and then infer the behavioural strategies in male frogs.

## 2. Results

To investigate interaction mechanisms inherent in the acoustic communication of actual animals, we analysed empirical data of male Japanese tree frogs that were obtained from our previous laboratory experiment and data analysis [19,26] (see §4.1 for details). In each trial of the experiment, we randomly captured three male frogs at a field site, and then placed them along a straight line at intervals of 50 cm between nearest neighbours. Spontaneous calling behaviour of the three frogs was recorded by three microphones that were placed in the vicinity of each frog. The audio data were separated into call signals of individual frogs using independent component analysis. Here, we used four datasets of the audio data in which each frog called more than 1400 times in four hours, allowing us to precisely estimate the parameters of a phase oscillator model by using the large sample size of call timing.

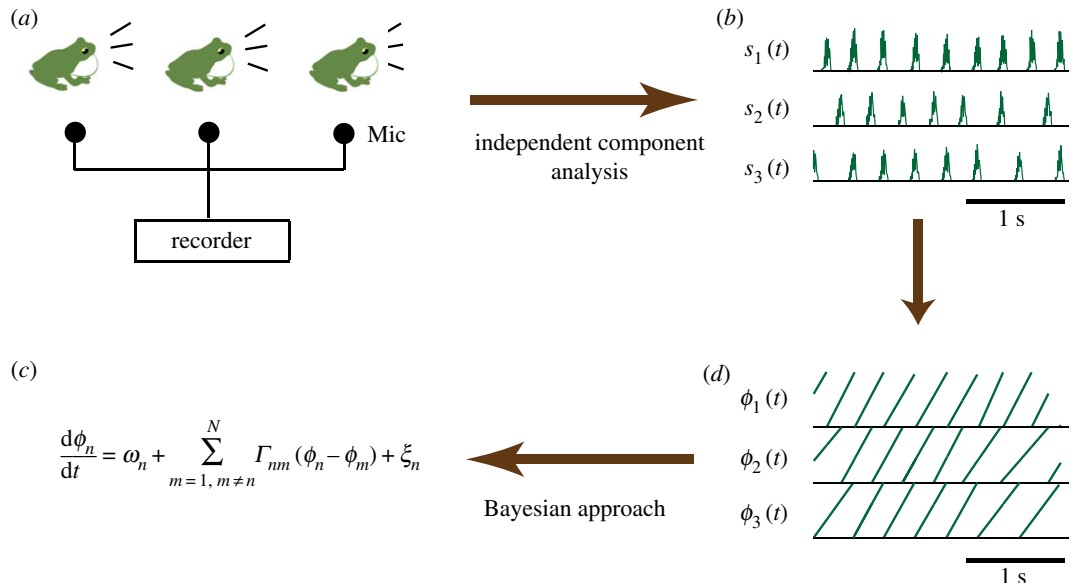

**Figure 2.** Schematic diagram on the identification of a phase oscillator model. In this study, we used the audio data of male Japanese tree frogs obtained from our previous study [19]. Phase dynamics was estimated from the audio data. We then estimated the parameters of a phase oscillator model by analysing the phase dynamics using a Bayesian approach, which allows us to infer the interaction mechanisms among the actual frogs. (*a*) Experiment, (*b*) call signals, (*c*) model identification and (*d*) phase dynamics.

Figure 2 explains how we estimate the parameters of a phase oscillator model with a general interaction term from the empirical data. A phase oscillator model is a well-known mathematical model that is theoretically derived from general classes of coupled oscillators [16] (see §4.2 for the details of a phase oscillator model). Experimental and theoretical studies have shown that the phase oscillator model with a simple interaction term can qualitatively reproduce the synchronization phenomena in various types of actual biological oscillators [27–30] including the chorus structures of male Japanese tree frogs [18–20,26]. In this study, we first calculated a phase $\phi_n(t_i)$ ($n = 1, 2, 3$) with discrete time $t_i$ according to equation (4.1) using the call timing of actual frogs (see §4.1 for details). Then, a Bayesian method [31] was applied to the time-series data of $\phi_n(t_i)$ so as to estimate the parameters of a phase oscillator model (i.e. an interaction term $\Gamma_{nm}(\phi_n(t_i) - \phi_m(t_i))$, a natural frequency $\omega_n$ and a noise intensity $\sigma_n$). In the following part, we first explain the result with a single dataset in detail (figures 3–6), and then comprehensively examine the relationship between the identified model and behavioural parameters using all the datasets (figure 7).

Figure 3 shows a representative result of the model identification. Cyan regions represent the interaction terms with the 95% confidence interval. To confirm the validity of the model identification, we performed a numerical simulation by using the identified model and compared it with empirical data. Note that we set only one parameter $\sigma_n$ to be slightly larger than its estimated value throughout the following analysis because this parameter was very likely to be estimated at a smaller value because of the intermittency of the chorus over a long time scale (see §4.1 for details) that is incompatible with the phase oscillator model. Figure 4*a* represents the scatter plot of phase differences that were obtained from numerical simulation of the identified model. By contrast, figure 4*b* shows the scatter plot of phase differences that were directly calculated from our empirical data. Here, we plot the phase differences only when one of the phases hits 0, which is consistent with our method for calculating a phase difference from discrete call timing [19,21]. The comparison of figure 4*a,b* demonstrates that the identified model can qualitatively reproduce the experimental result, supporting the validity of our model identification.

In addition, our empirical data of figure 4*b* show a complex transition among clustered anti-phase synchronization and tri-phase synchronization, which is consistent with our previous study [19]. To investigate the mechanism of such a transition, we examined the existence of an equilibrium state (a stable equilibrium state) and a critical state in the identified model. It is known that both of the states can reproduce synchronized features in dynamical systems [27] while their stabilities are different from each other. For instance, an equilibrium state with two oscillators gives the situation in which

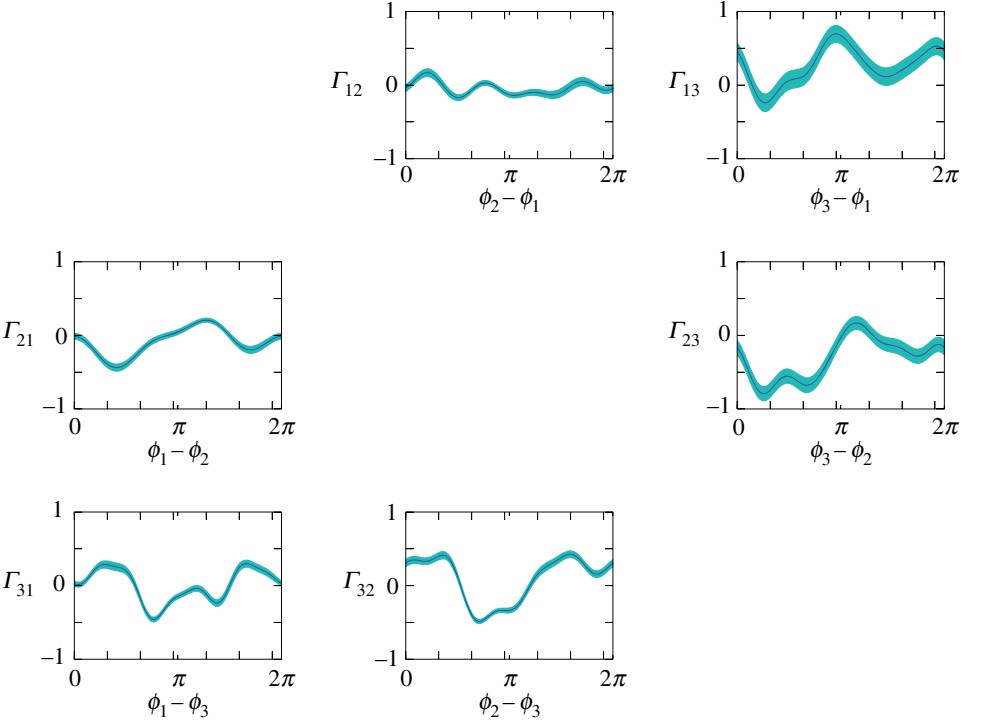

**Figure 3.** Unidirectional interaction terms of a phase oscillator model that were identified from the empirical data by a Bayesian approach. In this study, the interaction term $\Gamma_{nm}$ describes how the $n$th frog controls its call timing by responding to the calls of the $m$th frog. Cyan region represents the 95% confidence interval of the identified interaction term.

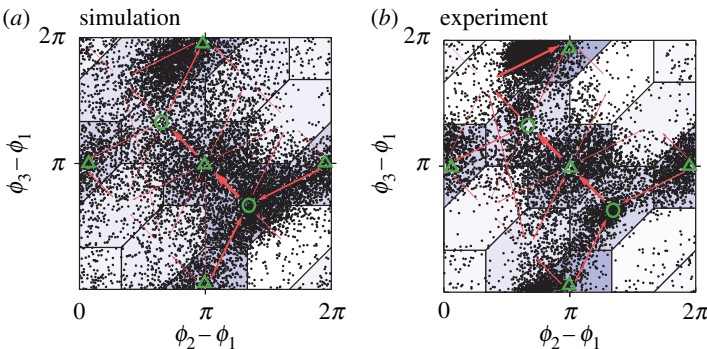

**Figure 4.** Phase differences between the calls of three male frogs that were obtained from (*a*) numerical simulation of the identified model and (*b*) behavioural experiment with actual frogs. Each black dot represents a set of the phase differences $\phi_2 - \phi_1$ and $\phi_3 - \phi_1$. Circle and triangle depict the regions of tri-phase synchronization and clustered anti-phase synchronization, respectively. Red arrows represent the transitions among the synchronization states. The width of each arrow is proportional to the number of the transitions.

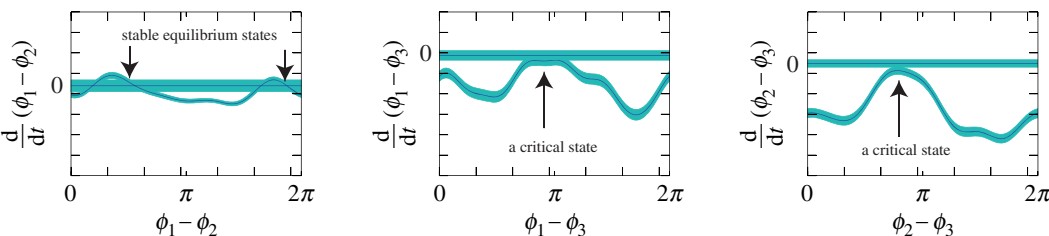

**Figure 5.** Critical states of the identified model. Cyan region represents the 95% confidence interval of $d(\phi_n - \phi_m)/dt$ that was estimated from the empirical data by a Bayesian approach. It is demonstrated that $d(\phi_1 - \phi_3)/dt$ and $d(\phi_2 - \phi_3)/dt$ have critical states while $d(\phi_1 - \phi_2)/dt$ has equilibrium states.

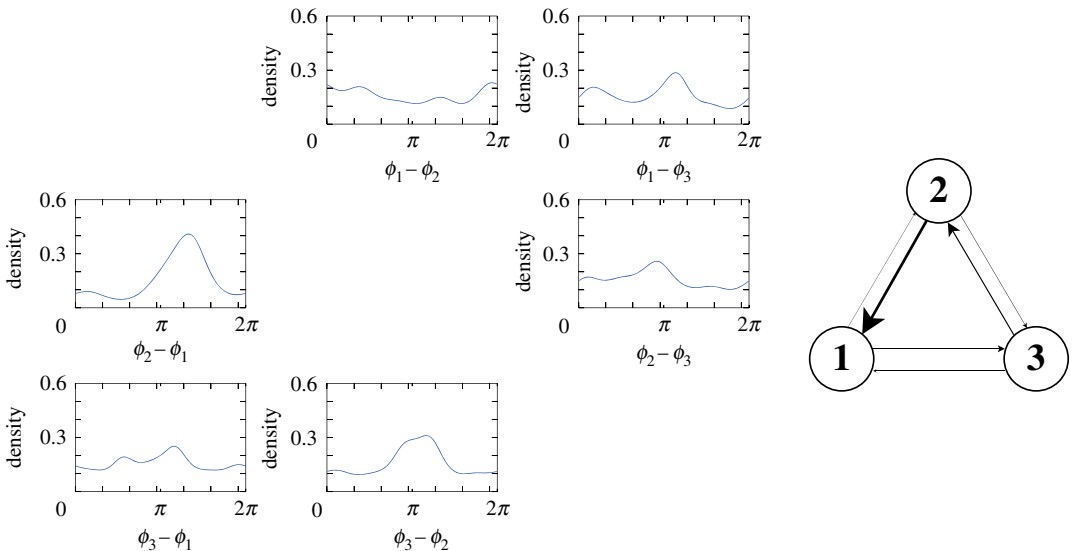

**Figure 6.** Selective attention quantified from the identified model. (Left) Stationary distribution of the phase differences that was obtained from the Fokker-Plank equation of the identified model. (Right) Schematic diagram of selective attention that was quantified by using the Kullback–Leibler divergence of the stationary distribution from uniform distribution. Line width represents the magnitude of attention paid among the male frogs.

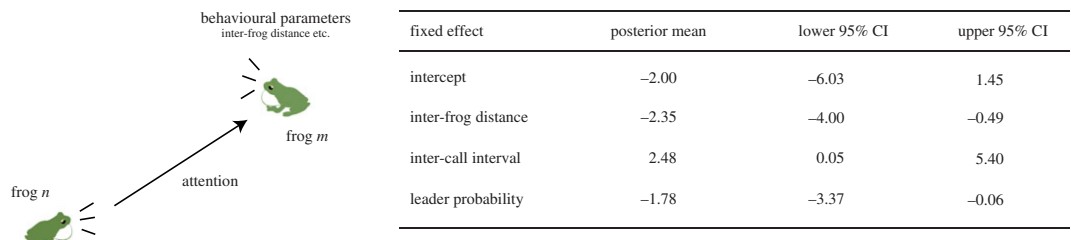

| fixed effect | posterior mean | lower 95% CI | upper 95% CI |
|---|---|---|---|
| intercept | −2.00 | −6.03 | 1.45 |
| inter-frog distance | −2.35 | −4.00 | −0.49 |
| inter-call interval | 2.48 | 0.05 | 5.40 |
| leader probability | −1.78 | −3.37 | −0.06 |

**Figure 7.** Relationship between selective attention and behavioural parameters examined by a statistical model (GLMM). The magnitude of attention is treated as a response variable; three behavioural parameters (i.e. inter-frog distance, inter-call interval and leader probability) are treated as fixed factors. This result was obtained from the empirical data that consist of four datasets with 12 frogs (the empirical data of call timing are available from electronic supplementary material of this manuscript for all the datasets).

the phase difference of the oscillators converges to a specific value (see electronic supplementary material, figure S2A for details), corresponding to the behaviour that two frogs produce calls at a specific interval quite robustly. By contrast, a critical state with two oscillators gives the situation in which the phase difference stays around a specific value for a long time and then intermittently leaves the value (see electronic supplementary material, figure S2B for details), corresponding to the behaviour that two frogs produce calls at a specific interval for a long time and then intermittently produce calls at an unspecific interval. To evaluate the existence of the two states, we analysed the relationship between a phase difference $\phi_n - \phi_m$ and its differential $\mathrm{d}(\phi_n - \phi_m)/\mathrm{d}t$ using the identified model (see §4.4). Note that the differential $\mathrm{d}(\phi_n - \phi_m)/\mathrm{d}t$ can be calculated from the right-hand side of equation (4.6). Figure 5 demonstrates that $\mathrm{d}(\phi_1 - \phi_3)/\mathrm{d}t$ and $\mathrm{d}(\phi_2 - \phi_3)/\mathrm{d}t$ have critical states while $\mathrm{d}(\phi_1 - \phi_2)/\mathrm{d}t$ has equilibrium states. We speculate that the existence of the critical states is the basis of the transition shown in figure 4 because (i) the stability of critical states is weaker than that of equilibrium states as explained above, and then (ii) the critical states are more sensitive to added noise.

Next, we quantified the magnitude of the attention paid among the male frogs based on the result of the model identification. The stationary distribution of a phase difference $\psi_{nm} \equiv \phi_n - \phi_m$ was first calculated as $\hat{f}(\psi_{nm})$ by numerically solving the Fokker–Planck equation of the identified model until it had converged (see §4.5). It should be noted that the stationary distribution $\hat{f}(\psi_{nm})$ gives the distribution of a phase difference that is most expected to be realized by the identified phase oscillator model. For instance, if the distribution $\hat{f}(\psi_{nm})$ has a sharp peak around $\pi$, it is likely that the $n$th frog attempted to call alternately with the $m$th frog. Figure 6 demonstrates that some distributions have a

sharp peak around $\pi$, indicating that some males attempted to call alternately with a specific male. Then, we calculated the Kullback–Leibler divergence of the stationary distribution $\hat{f}(\psi_{nm})$ from uniform distribution for each pair of the male frogs (see §4.5), which corresponds to the quantification of the attention. The right panel of figure 6 shows the result of the quantification in which line width represents the magnitude of the attention. Consequently, an asymmetric structure is observed in the attention paid among the male frogs. For example, it is indicated that the first frog paid strong attention to the second frog while the second frog paid just weak attention to the first frog.

To further examine the validity of the model identification, we analysed the relationship between attention and behavioural parameters using a generalized linear mixed model (see §4.6). In this analysis, the magnitude of the attention is treated as a response variable, and three behavioural parameters (i.e. inter-frog distance, the probability of being a chorus leader (leader probability), and inter-call interval) are treated as fixed factors. These variables were calculated from four datasets with 12 frogs in which each frog stably produced calls (see §4.1 and electronic supplementary material). Figure 7 shows the posterior mean and the 95% confidence interval of the coefficients of respective fixed factors. This analysis demonstrates that (i) inter-frog distance has a negative effect on attention, meaning that male frogs paid more attention to their closer neighbour, and also indicates that (ii) leader probability and inter-call interval have negative and positive effects on attention, respectively. It should be noted that the 95% confidence intervals of the coefficients of inter-call interval and leader probability do not include but are very close to 0 while the 95% confidence interval of the coefficient of inter-frog distance is obviously below 0 (figure 7). This suggests that the effects of inter-call interval and leader probability on the attention are significant but are relatively marginal compared to the effect of inter-frog distance.

## 3. Discussion

In this study, we quantified interaction mechanisms in the chorus of male Japanese tree frogs by estimating the parameters of a phase oscillator model from empirical data. The identified model qualitatively reproduced the stationary and dynamical features of the frog choruses, supporting the validity of our model identification. Then, the magnitude of the attention paid among the male frogs was quantified on the basis of the identified model. To our knowledge, this is the first study that shows the evidence of selective attention inherent in an animal chorus by combining empirical data with the phase oscillator model.

The relationship between the attention and the behavioural parameters (figure 7) gives perspectives on the choruses of male Japanese tree frogs.

— **Inter-frog distance.** Our analysis using a statistical model demonstrates the negative relationship between attention and inter-frog distance, which means that male Japanese tree frogs pay more attention to their closer neighbour. This is consistent with previous studies reporting that a neighbouring pair of males alternate their calls in various species of frogs [4,5,12,21,32]. Because sounds attenuate depending on distance, the calls of a neighbouring pair should arrive at a higher intensity than the calls of a distant pair. We speculate that such a sound attenuation depending on distance is the basis of the negative relationship between attention and inter-frog distance.

— **Inter-call interval.** In various species of frogs, females prefer a conspecific male that produces calls at a higher repetition rate [33]. Our analysis indicates that male Japanese tree frogs pay more attention to a male that produces calls at a lower repetition rate (i.e. a longer inter-call interval). This result suggests that the male frogs pay more attention to a less attractive male and then call alternately with him. Given that alternating chorus patterns can reduce the acoustic interference of their calls [18,23], this feature would be important for male frogs to effectively advertise themselves to conspecific females by using the advantage of their attractiveness over a neighbouring male.

— **Leader probability.** Our analysis indicates that male Japanese tree frogs pay more attention to a male that rarely leads other males. In this study, we define a chorus leader as a male that started calling earlier than other males in the same chorusing bout (see §4.6 and also electronic supplementary material, figure S3). This is because robust formation of chorusing bouts are characteristic to male Japanese tree frogs [26]. However, it should be noted that our definition on the leader–follower relationship is related to but is different from a traditional definition. Namely, a chorus leader is traditionally defined on the basis of relationship between adjacent calls of male frogs. For instance, if Frog 1 partially overlaps a call with Frog 2 but produces the call slightly earlier than Frog 2, Frog 1 is defined as a chorus leader [34,35]. In the context of the traditional definition, it is known that female

frogs generally prefer a chorus leader rather than a chorus follower because the onset of calls of the leader is not masked by calls of the follower [34,35]. Given that, based on our definition of a chorus leader, a leading male can also avoid masking of his calls by those of followers at the beginning of a chorusing bout (see electronic supplementary material, figure S3), we speculate that our result on negative correlation between attention and leader–follower relationship indicates that a male frog paid attention to less attractive male frogs (i.e. chorus followers). However, further behavioural experiments using female *H. japonica* are required because an acoustic preference for chorus leaders (our definition) has not yet been examined.

Thus, our analysis is likely to show the important features in frog choruses relevant to acoustic communication and mating. However, it should be noted that Neelon and Höbel reported an inconsistent result with our indication. Namely, their playback experiment demonstrated that male frogs (*H. cinerea*) selectively attend the calls of more attractive males [14]. Given that they worked on other frog species and used other call traits (i.e. sound frequency) as an indicator of attractiveness, comprehensive studies on various frog species would be required to further examine the attention of chorusing males in the context of their mating strategy.

The application of our methodology to a variety of empirical data would contribute to further understanding of selective attention in frog choruses. In our laboratory experiment, we used a linear arrangement of subjects because male Japanese tree frogs are often positioned along edges of a paddy field [20]. Therefore, this was a reasonable approximation of the actual spatial distribution of the male frogs at a field site. However, unlike the evenly spaced males in our laboratory experiment, inter-frog distance can vary among linearly arranged males at paddy fields. With respect to this point, even with a non-even distribution of Japanese tree frogs, we observed that each pair of nearest neighbours tends to alternate their calls in the field [21]. Nevertheless, because the spatial distribution of male frogs in two and three dimensions can vary among species and chorusing venues [4], additional investigation of call timing among males under a variety of distribution patterns is certainly warranted. Given that the distance among male frogs profoundly affects the loudness of calls that other frogs recognize, the magnitude of attention is probably affected by the spatial distribution of male frogs. Related to this point, empirical studies indicate further complexity in selective attention. For instance, Schwartz reported that male *H. microcephala* would adjust call timing in response to more than just the calls of their nearest neighbour [15]; Schwartz *et al.* reported that selective attention among nearest neighbours is not observed in male *H. versicolor* when more than two males call [36]. Thus, the features of selective attention vary a lot depending on species, and therefore further studies are required to comprehensively examine the mechanisms of selective attention in frog choruses.

The present methodology is widely applicable to the analysis on various types of communication in animals because (i) a phase oscillator model is derived from simple assumptions about periodicity and interaction, both of which are common in the communication of animals relying on various signals such as sounds [4,5], lights [37,38], visual display [39,40] and electric fields [41], (ii) our methodology only requires the timing of signal emissions to estimate the parameters of a phase oscillator model, and (iii) our methodology allows us to separately identify unidirectional interaction terms in a phase oscillator model. An important point is that the interaction term of the phase oscillator model varies depending on the value of the phase difference that corresponds to the change in the inter-signal interval among individual animals. This study demonstrates that such a dynamical property of the phase oscillator model can precisely capture not only the stationary distribution of the phase difference but also the dynamical feature of the transition among multiple synchronization states. We believe that this property of a dynamical model is advantageous for the analysis on selective attention compared to traditional methods (e.g. the calculation of the histogram of the inter-signal intervals [4,42]) especially when animal communication shows complicated dynamics.

On the other hand, a technical aspect of our methodology needs to be addressed. In this study, we estimated a phase from empirical data using a piecewise linear function (see equation (4.1)). Namely, we defined a call with $\phi_n = 0$, and then interpolated the time evolution of the phase as linearly increasing to the next call. Because we could not directly observe the phase from empirical data, this procedure was necessary but could be a source of observation error. Note that a similar issue occurs in other biological oscillators such as spiking neurons [43]. Given that the phase oscillator model identified by our method succeeded in qualitatively reproducing the distribution of phase differences, we speculate that the noise term of a phase oscillator model (i.e. $\xi_n(t)$ in equation (4.2)) can effectively incorporate the noise component including the observation error in the case of frog choruses. However, further studies are required to clarify the validity and limitations of our methodology, comparing the performance of other relevant methods [43,44].

# 4. Material and methods

## 4.1. Estimation of phase dynamics from empirical data

In this study, we use the empirical data of male Japanese tree frogs that were obtained from our previous laboratory experiment [19]. For each trial of the experiment, we performed the following procedures:

— Three individuals of male Japanese tree frogs were captured at a paddy field at Kyoto University (3501′57.2″ N, 13 547′00.0″ E).
— Each frog was put in a small mesh cage, and then three cages were placed along a straight line at intervals of 50 cm between nearest neighbours.
— We placed three microphones (Sony, ECM-55B) in the vicinity of each cage, and recorded spontaneous calling behaviour of the three frogs at sampling rate of 48 kHz using the three microphones and a recorder (Roland, R-4 PRO).
— After the audio recording, the male frogs were released at the same paddy field where they had been captured.

These procedures were carried out 44 times between 2008 and 2009. In most trials, some of the three frogs rarely called or did not call at all. Because this study aims to estimate the parameters of a mathematical model using a statistical approach and the precision of the parameter estimation depends on a sample size of the empirical data, we chose four datasets in which each male produced calls more than 1400 times in four hours. The four datasets were obtained from the trials performed on 26 May, 16 June, 17 June in 2008 and 29 May in 2009. The audio data of each of the four datasets were separated into call signals of individual frogs using independent component analysis [19]. In addition, we carefully checked the quality of the sound-source separation and excluded just three chorusing bouts from 143 chorusing bouts in which the separation did not work well [26]. From the audio data, we estimated the call timing of frogs according to the method of [19] (all the datasets of the call timing are available in the electronic supplementary material). The laboratory experiment and collection of Japanese tree frogs were carried out within the facility of Kyoto University in accordance with the guidelines approved by the Animal Experimental Committee of Kyoto University.

For each dataset, the call timing of three male frogs is described as $T_{n,k}$ that represents the timing of the $k$th call vocalized by the $n$th frog ($n = 1, 2$ or $3$). By using the call timing $T_{n,k}$, we describe a phase $\phi_n(t_i)$ for the $n$th frog at discrete time $t_i$ as follows [19,20,45]:

$$\phi_n(t_i) = 2\pi \frac{t_i - T_{n,k}}{T_{n,k+1} - T_{n,k}}. \tag{4.1}$$

Here, the phase $\phi_n(t_i)$ linearly increases from 0 to $2\pi$, and then is reset to 0 when the $n$th frog produces a call. Subsequently, we can interpolate the phase of the $n$th frog even when he does not produce a call, by substituting call timings of other frogs into $t_i$. To calculate the phase $\phi_n(t_i)$, we only consider the cases that successive call timings $T_{n,k}$ and $T_{n,k+1}$ satisfy the conditions of $T_{n,k} \leq t_i < T_{n,k+1}$ and $T_{n,k+1} - T_{n,k} < 0.9$ s. The first condition (i.e. $T_{n,k} \leq t_i < T_{n,k+1}$) is necessary for restricting the phase to the range of $0 \leq \phi_n(t_i) < 2\pi$. The second condition (i.e. $T_{n,k+1} - T_{n,k} < 0.9$ s) is necessary because male Japanese tree frogs intermittently start and stop their periodic calling behaviour over a long time scale. Namely, each frog periodically produces calls nearly every 0.3 s for several tens of seconds, stays silent for several minutes, and then repeats this cycle [26]. Because such an intermittency over a long time scale is incompatible with the phase oscillator model, we use the second condition in order to calculate the phase $\phi_n(t_i)$ only during the periodic calling behaviour.

## 4.2. Phase oscillator model

To reproduce the stationary and dynamical features in the choruses of male Japanese tree frogs, we use a phase oscillator model [16] with additive noise as follows:

$$\frac{\mathrm{d}\phi_n(t)}{\mathrm{d}t} = \omega_n + \sum_{m=1,m \neq n}^{N} \Gamma_{nm}(\phi_n(t) - \phi_m(t)) + \xi_n(t), \tag{4.2}$$

where $\phi_n(t) \in [0, 2\pi)$ ($n = 1, 2, \ldots, N$) is the phase of the $n$th frog, and $\omega_n$ is a positive parameter that describes the intrinsic angular velocity of the $n$th frog. We then assume that the $n$th frog produces a call at $\phi_n(t) = 0$,

which is consistent with the definition of equation (4.1). Consequently, $2\pi/\omega_n$ gives the intrinsic inter-call interval of the $n$th frog [18–20,26]. $\Gamma_{nm}(\phi_n(t) - \phi_m(t))$ is the unidirectional interaction term between the $n$th frog and the $m$th frog, and is defined as a $2\pi$-periodic function of a phase difference $\phi_n(t) - \phi_m(t)$ [16]. In the context of acoustic communication among male frogs, this term represents how the $n$th frog controls his call timing by responding to the calls of the $m$th frog [18–20]. $\xi_n(t)$ is the term for additive noise. We assume that this term is given by independent white Gaussian noise with magnitude $\sigma_n$ that satisfies the relationship $< \xi_n(t)\xi_n(s) > = \sigma_n\delta(t-s)$. Note that we use this term $\xi_n(t)$ so as to better fit the model parameters from empirical data that contain a noise component. The terminology used in the phase oscillator model is summarized in electronic supplementary material, table S1.

## 4.3. Identification of a phase oscillator model

Given the time-series data of the phase $\phi_n(t_i)$ that was obtained from our experiment using actual frogs (see §4.1), we estimated the unknown parameters of the phase oscillator model according to the Bayesian method of [31]. In the method, we first define the following likelihood function from the time-series data of the phase $\phi_n(t_i)$:

$$L_n = \prod_i \mathcal{N}\left[ \frac{\phi_n(t_{i+1}) - \phi_n(t_i)}{t_{i+1} - t_i} \middle| \omega_n + \sum_{m=1,m \neq n}^{3} \Gamma_{nm}(\psi_{nm}(t_i)), \frac{\sigma_n^2}{t_{i+1} - t_i} \right],$$

(4.3)

where $\mathcal{N}(x|m, s^2)$ represents a Gaussian function with mean $m$ and variance $s^2$, and $\psi_{nm}$ denotes the phase difference $\phi_n - \phi_m$. Because each interaction term is defined as a $2\pi$-periodic function of the phase difference $\psi_{nm}$ [16], it can be expanded into a Fourier series as follows:

$$\Gamma_{nm}(\psi_{nm}) = \sum_{k=1}^{M} (a_{nm}^{(k)} \cos k\psi_{nm} + b_{nm}^{(k)} \sin k\psi_{nm}).$$

(4.4)

We applied a standard model comparison method to determine the maximum order $M$ in equation (4.4). Namely, we chose $M$ such that a marginal-likelihood function $\mathcal{L}_M$ is maximized. Technically, we computed and compared $\mathcal{L}_M$ for $M = 1, 2, \ldots, 30$. For our data, the maximum order $M$ was chosen between 1 and 9.

Then, the parameters to be estimated are $\omega_n, a_{nm}^{(1)}, \ldots, a_{nm}^{(M)}, b_{nm}^{(1)}, \ldots, b_{nm}^{(M)}$ and $\sigma_n$, which are denoted by a shorthand notation $c_n$. To estimate $c_n$ in a Bayesian framework, we use a reasonable conjugate prior distribution $p_{\text{prior}}(c_n)$, which is a Gaussian-inverse-gamma function. Bayes' theorem then gives the posterior parameter distribution as follows:

$$p_{\text{post}}(c_n) \propto L_n(c_i)p_{\text{prior}}(c_n).$$

(4.5)

Because the prior distribution $p_{\text{prior}}(c_n)$ is conjugate to the likelihood $L_n(c_i)$, we can easily calculate the posterior distribution $p_{\text{post}}(c_n)$ (see [31] and its electronic supplementary material for details).

## 4.4. Examination of critical states

Empirical data on the choruses of male Japanese tree frogs demonstrate the complicated transition among multiple synchronization states (figure 4b) [19]. Here, we examine the origin of this transition on the basis of the identified model. Equation (4.2) without a noise term yields the time evolution of the phase difference $\psi_{nm} \equiv \phi_n - \phi_m$ as follows:

$$G_{nm}(\psi_{nm}) \equiv \frac{d(\phi_n - \phi_m)}{dt} = \omega_n - \omega_m + \Gamma_{nm}(\psi_{nm}) - \Gamma_{mn}(-\psi_{nm}),$$

(4.6)

where $\Gamma_{nm}(\psi_{nm})$ is the maximum *a posteriori* (MAP) estimation of the interaction term that was obtained from the analysis of §4.3. This function $G_{nm}(\psi_{nm})$ quantifies how the phase difference changes in time under the mutual interaction between the $n$th and $m$th frogs without the effect of another frog. If $G_{nm}(\psi_{nm})$ has a zero-crossing (a stable equilibrium state) at $\psi_{nm} = \psi_{nm}^*$, the phase difference approaches $\psi_{nm}^*$ and then stays near the value forever (see also electronic supplementary material, figure S2A). Such stationary dynamics typically come from strong interaction. By contrast, if $G_{nm}(\psi_{nm})$ has no zero-crossing but is very close to zero at a certain point $\psi_{nm} = \psi_{nm}^{**}$ (a critical state), the phase difference stays near $\psi_{nm}^{**}$ for a long time and intermittently departs the point (see also electronic supplementary material, figure S2B). Such critical dynamics typically come from moderate interaction. The

terminology used in the time differential equation of the phase difference (equation (4.6)) is summarized in electronic supplementary material, table S2.

## 4.5. Quantification of selective attention

To assess selective attention among male frogs, we evaluated the stochastic feature of the identified model by taking the effect of noise into consideration. For example, to evaluate the attention from the $n$th frog to the $m$th frog, we analysed the following model only with the unidirectional interaction term $\Gamma_{nm}(\phi_n(t) - \phi_m(t))$:

$$\frac{\mathrm{d}\phi_n}{\mathrm{d}t} = \omega_n + \Gamma_{nm}(\phi_n(t) - \phi_m(t)) + \xi_n(t) \tag{4.7}$$

and

$$\frac{\mathrm{d}\phi_m}{\mathrm{d}t} = \omega_m + \xi_m(t). \tag{4.8}$$

These equations describe the situation that the $n$th frog exposed to the noise term $\xi_n(t)$ pays attention to the $m$th frog according to the interaction term $\Gamma_{nm}(\phi_n(t) - \phi_m(t))$ (equation (4.7)) while the $m$th frog exposed to the noise term $\xi_m(t)$ does not pay any attention to the $n$th frog (equation (4.8)). Hence, this is a concise mathematical model capturing the attention from the $n$th frog to the $m$th frog under the effect of the noise $\xi_n(t)$ and $\xi_m(t)$. Subtracting equation (4.8) from equation (4.7) yields the time evolution of the phase difference $\psi_{nm} \equiv \phi_n(t) - \phi_m(t)$ as follows:

$$\frac{\mathrm{d}\psi_{nm}}{\mathrm{d}t} = \omega_n - \omega_m + \Gamma_{nm}(\psi_{nm}) + \xi_n(t) + \xi_m(t). \tag{4.9}$$

Then, the Fokker–Planck equation of equation (4.9) is given as follows:

$$\frac{\partial f(\psi_{nm}, t)}{\partial t} = -\frac{\mathrm{d}\Gamma_{nm}}{\mathrm{d}\psi_{nm}}(\psi_{nm})f - [\omega_n - \omega_m + \Gamma_{nm}(\psi_{nm})]\frac{\partial f}{\partial \psi_{nm}} + \frac{\sigma_n^2 + \sigma_m^2}{2}\frac{\partial^2 f}{\partial \psi_{nm}^2}, \tag{4.10}$$

This equation governs the time evolution of $f(\psi_{nm}, t)$ that represents the distribution of the phase difference $\psi_{nm}$ at time $t$. Subsequently, we can calculate the stationary distribution of the phase difference by numerically solving equation (4.10) until $f(\psi_{nm}, t)$ converges, and then describe the stationary distribution as $\hat{f}(\psi_{nm})$. It should be noted that the stationary distribution $\hat{f}(\psi_{nm})$ gives the distribution of a phase difference that is most expected to be realized by the identified phase oscillator model, allowing us to quantify the degree of attention. For instance, if the stationary distribution $\hat{f}(\psi_{nm})$ is almost uniform, it is likely that the $n$th frog did not pay any attention to the $m$th frog; on the other hand, if the distribution has a sharp peak around $\pi$, it is likely that the $n$th frog attempted to alternate calls with the $m$th frog. In this study, we quantify such a sharpness of the distribution $\hat{f}(\psi_{nm})$ using the Kullback–Leibler divergence from uniform distribution $u(\psi_{nm}) \equiv 1/2\pi$ as follows:

$$D_{\mathrm{KL}}(\hat{f}\|u) = \int_0^{2\pi} \hat{f}(\psi_{nm})\log\frac{\hat{f}(\psi_{nm})}{u(\psi_{nm})}\,\mathrm{d}\psi_{nm}. \tag{4.11}$$

Consequently, $D_{\mathrm{KL}}(\hat{f}\|u) \sim 0$ indicates no attention from the $n$th frog to the $m$th frog while $D_{\mathrm{KL}}(\hat{f}\|u) \gg 0$ indicates strong attention from the $n$th frog to the $m$th frog. The terminology related to the Fokker–Plank equation is summarized in electronic supplementary material, table S3.

## 4.6. Relationship between selective attention and behavioural parameters

To further examine the validity of the model identification, we analysed the relationship between selective attention and behavioural parameters. Here, we focus on the following behavioural parameters of male Japanese tree frogs: (i) the inter-frog distance, (ii) the inter-call interval, and (iii) the leader–follower relationship. The inter-frog distance represents the distance between each pair of male frogs that was measured in our experiments. Because three frogs were deployed along a line at intervals of 50 cm between nearest neighbours in our laboratory experiment (see §4.1), the inter-frog distance varied between 50 and 100 cm depending on pairs of male frogs (i.e. the distance between nearest neighbours was 50 cm while the distance between a distant pair was 100 cm). Then, the inter-call interval was calculated as $\delta T_{n,k} = T_{n,k+1} - T_{n,k}$ using the sequences of the call timing $T_{n,k}$ only when the condition $T_{n,k+1} - T_{n,k} \leq 0.6$ s is satisfied. The leader–follower relationship was determined

according to the following definition: the leader, the first follower, and the second follower are defined as the frogs that start calling first, second and third within the same chorusing bout, respectively (see electronic supplementary material, figure S3). Note that male Japanese tree frogs start calling with low-intensity sound, making it difficult for us to automatically determine the leader–follower relationship. Hence, we manually determined the leader–follower relationship by carefully looking at all the separated audio data and calculated the probability of making each frog the chorus leader.

Next, we analysed the relationship between the magnitude of attention and the three behavioural parameters using a generalized linear mixed model (GLMM). GLMM is a well-known statistical model that is used in various research areas to analyse the effects of multiple explanatory variables on a response variable [46,47]. Here, we treat the magnitude of the attention paid from the $n$th frog to the $m$th frog (see equation (4.11)) as a response variable $Y_{\text{att}}$. We then treat the three behavioural parameters (i.e. the inter-frog distance between the $n$th and $m$th frogs, the inter-call interval of the $m$th frog, and the leader probability of the $m$th frog) as fixed factors, and describe them as $X_{\text{dis}}$, $X_{\text{int}}$ and $X_{\text{prob}}$, respectively. In addition, we treat frog index and experimental date as random factors ($\xi_{\text{frog}}$ and $\xi_{\text{date}}$) because these factors are difficult to quantify but are very likely to affect the calling behaviour of male frogs. Here, we assume that the random factors $\xi_{\text{frog}}$ and $\xi_{\text{date}}$ follow a normal distribution of zero mean value with standard deviations of $\sigma_{\text{frog}}$ and $\sigma_{\text{date}}$, respectively. By using these variables, we construct the following GLMM

$$\log \alpha = \beta_0 + \beta_{\text{dis}}X_{\text{dis}} + \beta_{\text{int}}X_{\text{int}} + \beta_{\text{prob}}X_{\text{prob}} + \xi_{\text{frog}} + \xi_{\text{date}} \tag{4.12}$$

and

$$Y_{\text{att}} \sim \text{Gamma}(\alpha\beta, \beta). \tag{4.13}$$

Here, the response variable $Y_{\text{att}}$ is always positive because of its definition (see equation (4.11)). To reproduce the positive distribution of the response variable $Y_{\text{att}}$, we assume that (i) $Y_{\text{att}}$ follows a gamma distribution that always takes a positive value (equation (4.13)), and (ii) the parameter $\alpha$ of the gamma distribution that gives the mean of the distribution is linked to the fixed factors and random factors with a log function (equation (4.12)); this framework is consistent with a traditional gamma regression with multiple factors. We confirmed that there is no multicollinearity among any pairs of fixed factors (the absolute value of Pearson's correlation coefficient was less than 0.36). Posterior distributions of all the unknown parameters (i.e. $\beta_0$, $\beta_{\text{dis}}$, $\beta_{\text{int}}$, $\beta_{\text{prob}}$, $\beta$, $\sigma_{\text{date}}$ and $\sigma_{\text{site}}$) were estimated from MCMC samples generated by R statistical software (v. 3.4.2) and Stan (v. 2.17.2). Note that we normalized the fixed factors from 0 to 1 prior to the calculation of the MCMC samples, and confirmed the convergence of the MCMC samples using $\hat{R}$ with a threshold of 1.01 [48]. The posterior mean and the 95% confidence interval of $\beta_0$, $\beta_{\text{dis}}$, $\beta_{\text{int}}$ and $\beta_{\text{prob}}$ are shown in figure 7.

Data accessibility. In this study, we used the empirical data obtained from our previous study [19]. The time-series data of call timing are available from electronic supplementary material of this manuscript.

Authors' contributions. K.O., I.A and T.A. designed the research; K.O. and I.A. analysed the data; K.O. performed simulation; K.O., I.A. and T.A. wrote the paper. All authors gave final approval for publication.

Competing interests. We declare we have no competing interests.

Funding. This study was partially supported by JSPS Grant-in-Aid for Challenging Exploratory Research (grant no. 16K12396) and Grant-in-Aid for Young Scientists (grant no. 18K18005) to I.A.

Acknowledgements. We thank K. Aihara, H. G. Okuno, R. Takeda, T. Mizumoto, H. Awano, K. Itoyama and Y. Bando for their valuable comments on this study. We also appreciate two anonymous reviewers providing valuable comments that greatly help us to improve the quality of the manuscript.

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
