## [Reviewer comments · Royal Society Open Science]

Review History

RSOS-191693.R0 (Original submission)

Review form: Reviewer 1

Is the manuscript scientifically sound in its present form?

Yes

Are the interpretations and conclusions justified by the results?

Yes

Is the language acceptable?

Yes

Do you have any ethical concerns with this paper?

No

Have you any concerns about statistical analyses in this paper?

Yes

Recommendation?

Accept with minor revision (please list in comments)

Comments to the Author(s)

The authors investigate the interaction between acoustic signals of Japanese tree frogs. They adopt a phase-coupling model and estimate the coupling function from data. The results are interesting and relevant, and certainly, deserve publication in RSOS. However, an additional discussion of the approach would be appropriate. Indeed, the authors infer the phase dynamics model, while their observations are not the phases, but rather singular events (which can be described, in stochastic process theory terms, as a point process). The phase is reconstructed as piecewise linear function between the events, but this may potentially be a source of errors. A similar problem appears for reconstruction of interaction of neurons, if only the spikes are available. Recently, approaches for inferring phase dynamics from observations of spikes have been suggested [Physical Review E, 96, 012209, 2017], in these approaches, one, however, assumes coupling in a Winfree form, not in a Kuramoto-Daido form. I think, discussion of these issues would be appropriate.

Review form: Reviewer 2

Is the manuscript scientifically sound in its present form?

Yes

Are the interpretations and conclusions justified by the results?

Yes

Is the language acceptable?

No

Do you have any ethical concerns with this paper?

No

Have you any concerns about statistical analyses in this paper?

No

Recommendation?

Major revision is needed (please make suggestions in comments)

Comments to the Author(s)

The paper by Ota et al. addresses details of the calling dynamics within choruses of the Japanese treefrog, *Hyla japonica*. As has been the case with earlier publications from this group of researchers and their colleagues, the analysis employs sophisticated mathematics used to study dynamical systems. Of particular interest and importance is their result that the male treefrogs engage in selective attention within groups. That is, the call timing of individuals responds mostly to closer (and presumably) louder individuals and to a lesser degree to males further away. This finding is consistent with those from a handful of other studies of frogs and chorusing insects but is nevertheless valuable given the small number of such studies and the nature of the analytic approach of Ota et al. – which relies on an understanding of dynamical systems beyond that, I suspect, of most students of anuran communication. This ignorant group, unfortunately, includes me. Accordingly, although I have done my best to follow the math, I cannot judge that aspect of the paper.

The writing, although generally not bad given that the authors are Japanese, required some work on my part to improve the English. In particular, there were many statements that I found unclear and others which could benefit from less wordy prose. I also feel that the paper would be improved significantly if there was more explanation of some of the mathematics.

The findings are significant, and it would be nice for readers who are not mathematicians, engineers or physicists to be able to understand more of what particular aspects of the analysis mean and accomplish (and why they do so). Ota et al. don't neglect this entirely but additional help would be quite valuable to many in their audience. As I suggest below, the authors could effectively deal with the terminology issue by including in their paper (perhaps in an Appendix) a glossary. This could be in the form of a table or enclosed "box" of definitions also indicating how the items specifically relate to their system of vocally interacting frogs. I'll be frank. Reading the publications of Aihara and colleagues has sometimes been frustrating because the mathematics has often not been accompanied by sufficient explanation. The work is superb but it is a struggle to understand much of the content of their papers.

The spatial arrangement of males used in this study was quite simple (3 evenly spaced males in a line). Therefore, I also strongly recommend that the authors discuss limitations of their study and expectations under more realistic (i.e. natural) geometries. If data are available from some of their other studies or those published by other researchers, in their discussion section they should incorporate relevant details.

Note below that I have mentioned particular citations when I feel that additional or more appropriate research needs to be credited. A list of the references is provided at the end of my specific comments.

Specific Comments

Lines 14-26. Modify the abstract to read as follows.

"We employed empirical data from multiple recordings and a phase oscillator model to describe the deterministic and stochastic features of frog choruses in which male Japanese tree frogs attempt to avoid call overlap. The mathematical model with a general interaction term is identified using a Bayesian approach and it qualitatively reproduces the stationary and dynamical features of the empirical data. In addition, we quantify the magnitude of attention paid among the male frogs from the identified model, and then analyze the relationship between attention and behavioral parameters using a statistical approach. Our analysis demonstrates a negative correlation between attention and inter-frog distance and also suggests a behavioral strategy in which male frogs selectively attend to a less attractive male frog in order to more effectively advertise their superior relative attractiveness to females."

Line 28. Change "form of a swarm" to "aggregations". I have never seen or heard the term "swarm" applied to a group of birds or frogs. It is more applicable to flying insects.

Line 29 - 30. Change "flexible flock" to "flexible school or flock". Change "To maintain the flock, they need to synchronize their velocity and direction with each other" to "To maintain the group, members need to synchronize their velocity and direction".

Line 31. Change "also mate identification" to "also mate identification and attraction".

Line 32. Change "acoustic signals with each other" to "acoustic signals".

Line 34. Replace "a swarm" with "the aggregation".

Lines 34-35. Alternation may also facilitate quality assessment by receivers in addition to be an indicator of quality. Accordingly, I recommend that you change the sentence to read as follows. "Thus, synchronization and alternation are common in animal choruses, and can both facilitate assessment of signalers and indicate their quality".

Line 37. Change "target in a swarm" to "target in an aggregation".

Line 40. Change “during a prey capture” to during prey capture”.

Line 41. Change “roles of the selective attention” to “roles of selective attention”. You cite two reviews [4,5]. Here or elsewhere you should mention papers that deal specifically with selective attention in frogs such as Brush and Narins (1989), Schwartz (1993), Greenfield and Rand (2000), Neelon and Hoebel (2019). Note that Schwartz et al. (2002) found that gray treefrogs (*Hyla versicolor*) do not selectively avoid overlap with their nearest neighbors.

Line 42. Change “their interaction mechanisms and compare it with the behavior of an individual animal” to “interaction mechanisms and evaluate these with respect to the behavior of an isolated animal”.

Lines 43-45. I’m not sure that here and elsewhere the word “identifying” is the best one to describe what you did. Perhaps you should say “estimating the parameters of a phase oscillator model” to be more consistent with what you state in line 82. I’d change the sentence to read: “This study aims to quantify interaction mechanisms in choruses of male Japanese tree frogs (*Hyla japonica*) by estimating the parameters of a phase oscillator model [12] from empirical data.”

Lines 45. Change “and breed mainly at a paddy field from April to July [13]” to “and often breed in paddy fields from April to July [13]”.

Line 46. Delete the word “successive”.

Lines 47-48. Change from “demonstrate that they avoid call overlaps with each other in the forms of” to “demonstrated that they avoid overlapping calls through”.

Line 50. Here and elsewhere in your manuscript use the past tense when describing what has already been done. So, change “reveal” to “revealed”.

Lines 52-54. Change “Because the temporal overlap of acoustic signals generally mask the information included in each call, these alternating behavior would be important for male frogs to effectively advertise themselves towards a conspecific female [14–18]” to “Because the temporal overlap of acoustic signals can mask or degrade the information included in each call (Vélez et al., 2013), these alternating behaviors [14–18] can help male frogs more effectively advertise themselves to conspecific females [e.g. Schwartz, 1987].”

Are results of female choice tests available for *H. japonica*? If so, do they discriminate in favor of non-overlapping calls? Please include appropriate citations if these experiments have been conducted. You should also be aware that females of not all species prefer non-overlapping calls (see results with *H. crucifer* in Schwartz, 1987).

Line 56. Change “Japanese tree frogs from our empirical data according to a Bayesian approach” to “Japanese tree frogs based on our empirical data using a Bayesian approach”.

Line 58. Change “parameters is analyzed by a statistical approach” to “is statistically analyzed.”

Line 61. Change “choruses as the form of a” to “choruses within a”.

Lines 67-68. Change “along a straight line at the interval of 50 cm in a stationary room” to “along a straight line at intervals of 50 cm”.

Note I have no idea what you mean by “a stationary room”. Rooms do not normally move.

Lines 68-69. In your methods section, you should give the model of your microphones and recorder.

Line 70. Change “of respective frogs” to “of individual frogs”.

Line 84. So, do the results presented in figures 3-6 come from just a single data set? If so, you should make this clearer.

Lines 86-87. Change “perform numerical simulation by using the identified model and compare it with empirical data” to “performed a numerical simulation by using the identified model and compared it with empirical data”.

Line 99. Change “further analyze” to “further analyzed”.

Line 101. You need to explain what you mean by “critical states” in contrast to “equilibrium states”. I believe you are referring to your finding that a small perturbation (contributed by “added noise”) can result in an abrupt change in the form of synchronization when there are critical states present. As I mentioned above, perhaps you could most effectively deal with the terminology issue by including in your paper (perhaps in an Appendix) a table of definitions and how they specifically relate to your system of vocally interacting frogs.

Line 103. Change “this property about critical states” to “this property of critical states”. Change “origin” to “basis”.

Based on my understanding of “critical states” this argument (lines 101-103) seems circular.

Line 105. Change “we quantify” to “we quantified”.

Line 106. After some checking online, I’m assuming that you are referring to the stationary distribution discussed by Kuramoto in his model of oscillators. But you really need to explain “stationary distribution” and why it is relevant. This could go in the table that I suggested you include in your paper. I realize that you say a little more about stationary distributions in section 4.5. However, more information would be very helpful so that readers can understand why the presence and form of a peak in the distribution indicates the nature of the timing between a pair of frogs and selective attention.

Change “difference is” to “difference was”.

Line 109. Change “Then, we calculate” to “Then we calculated”.

Line 116. “we analyze the relationship between the attention” to “we analyzed the relationship between attention”.

Lines 119-120. Change “(i.e., an inter-frog distance, the probability for taking a chorus leader (leader probability), and an inter-call interval)” to “(i.e., inter-frog distance, the probability of being a chorus leader (leader probability), and inter-call interval)”.

Above, I hope I have interpreted what you meant by “taking” correctly.

Line 124. Change “on the attention” to “on attention”.

Line 125. Change “the leader probability”, “the inter-call interval” and “the attention” to “leader probability”, “inter-call interval” and “attention”.

Line 138. Change “to their neighbor” to their “closer neighbor”. Given the small distance between males in your experiment, both males could be said to be neighbors. Also from the data

in this paper, it is not clear what would happen when there are more than 3 males calling in a line. The Aihara et al. (2014) paper in Scientific Reports discusses this situation (at least when more males are calling) with data from your lab's Firefly system. In your introduction, you mention the two-cluster synchronization observed in that study. However, those data were obtained from males positioned linearly along the edge of a paddy field. I'm curious as to whether you have data from any field sites (or lab tests) with males scattered in a nonlinear pattern – as would likely occur in many natural calling sites.

Lines 141-142. Again, it is important that you briefly discuss the issue of non-linear and non-even spacing intervals between males. Such spatial distributions can lead to more complex patterns of call timing between chorusing males. Schwartz (1993), for example, found that some males of *H. microcephala* (located in more interior positions) would adjust note timing in response to more than just the notes of their loudest neighbor.

Line 145. I would change "leads their chorus" to "leads another". The studies referenced (27 and 28) used or referred to results of 2-speaker tests with females. It is not certain what would happen when more males are calling. Also, reference 28 was a study of males. A more appropriate reference would be (Greenfield and Roizen 1993).

Lines 147-148. You should check out the recent paper by Neelon and Hoebel (2019) and mention their finding that males selectively attend to the calls of more attractive males.

Line 150. Change "towards" to "to".

Lines 152-153. Change "Thus, our analysis is likely to show the important features in frog choruses, which is valid in the contexts of the acoustic communication of male frogs as well as the strategy for mating" to "Thus, our analysis is likely to show important features in frog choruses relevant to acoustic communication of frogs and mating".

Line 155. Change "but very close" to "but are very close".

Line 159. Change "because of the following reasons:" to "because:".

Line 161. Change "relying on various signals (e.g., sounds" to "relying on signals such as sounds".

Remember to delete the closed parenthesis ")" on the next line.

Line 162. Delete "so as".

Line 170-171. Change "inter-signal interval) when studying selective attention in animal communication that shows complicated dynamics" to "inter-signal intervals [Klump and Gerhardt, 1992]) when studying selective attention when animal communication shows complicated dynamics".

As I have indicated above, you should probably cite a reference here such as Klump and Gerhardt (1992).

Line 176. In "paddy field of Kyoto University" do you mean "at Kyoto University" or "belonging to Kyoto University" and located elsewhere?

Lines 178-179. Change "line at the interval" to "line at intervals".

As I indicated above, I'm not sure what you mean by "a stationary room". Most rooms are stationary. Do you mean that the positions of the caged frogs were stationary?

Line 180. Change "by three" to "using three". You should provide information on the type (manufacturers, models) of recording equipment.

Line 183-184. Change “44 times in total” to “44 times”. Change “trials of the experiment, some of three” to “trials, some of the three”.

Line 185. “Change “due to” to “using”.

Line 186-187. Change “on a sample size of the empirical data, we choose” to “on a sample size of the empirical data, we chose”.

Some clarification is also needed here. By “all the three frogs” I think you mean that the total number of calls given by the three frogs (calls of frog 1 + calls of frog 2 + calls of frog 3) was required to exceed 1400. Is this the case? Or, was each male required to give 1400 calls?

Begin a new sentence on line 187 with “The four datasets were”.

Line 188-190. Change “May in 2009, respectively. Then, the audio data of the four datasets were analyzed by the method of independent component analysis, and were separated into call signals of respective frogs [15]” to “May in 2009. The audio data of each of the four datasets were separated into call signals of individual frogs using independent component analysis [15].”

Line 191. Based on what you state in lines 200-201, I believe that you mean “three chorusing bouts from 143 chorusing bouts” rather than “three choruses from 143 choruses”. If so, make the necessary change.

Line 192. Delete the word “respective”.

Line 195. Change “guideline” to “guidelines”.

Line 199. Change “phase in the range” to “phase to the range”.

Line 201. Change “calls almost at the interval” to “calls nearly every”.

Line 203. If this is what you mean, make the following change. Substitute “incompatible with” for “out of the scope of”.

Line 205. Change “and then reset to be 0” to “and then is reset to 0”.

Line 214. Change “the term of” to “the term for”.

Line 223. Change “material for detail” to “material for details”.

Lines 241-242. Change “(1) an inter-frog distance, (2) an inter-call interval, and (3) leader-follower relation-ship. An inter-frog distance” to “(1) the inter-frog distance, (2) the inter-call interval, and (3) the leader-follower relation-ship. The inter-frog distance”

I’m curious. You always used a male-male separation of 50 cm. Have you varied the spacing so that different pairs of males are separated by different distances?

Line 247. Change “the same chorus” to “the same chorusing bout”.

Line 250. What you did is a little unclear to me. In any event, I think the following wording is better. If you agree, make the following change.

Substitute “of making each frog the chorus leader” for “for taking a chorus reader for each frog”. Note also that you misspelled “leader” as “reader”.

For the next paragraph, the continuous line numbering was absent. So, I will refer to the page-specific line numbers.

Page 13, line 20. Change “areas so as to” to “areas to”.

Page 13, line 26. Do you mean “and fixed factors” rather than “of fixed factors”? That is, do you mean that the variables you list listed are ‘fixed factors’ in your model?

Page 13, line 29. Should this be “variables and random factors” rather than “variables of random factors”?

Change “difficult to be quantified” to “difficult to quantify”.

Lines 251-252. You state “To reproduce this feature”. I’m not certain to what feature you are referring.

Line 257. Change “coefficient is less than” to “coefficient was less than”.

Line 311. For reference 18, change “begins wells” to “begins Wells”. Wells refers to the last name of Kentwood Wells and so should be capitalized.

Line 326. Change “bayesian” to “Bayesian”.

Figure Captions

For Figure 1C, males 1 and 3 do overlap. So you need to rephrase this.

Page 18. Line 53 (Caption of Figure 2). Change “dynamics according to” to “dynamics using a”.

Page 20. Line 29. Change “behavioral experiment of actual frogs” to “the behavioral experiment with actual frogs”.

Page 22. Lines 33-38. You may need to reword this figure caption based on changes you may need to make that I mention above for Page 13, line 26.

References that you should cite which are not already in your references section.

Greenfield, M. D. & Roizen, I. 1993 Katydid synchronous chorusing is an evolutionarily stable outcome of female choice. *Nature* 364, 618-620.

Greenfield MD, Rand AS (2000) Frogs have rules: Selective attention algorithms regulate chorusing in *Physalaemus pustulosus* (Leptodactylidae). *Ethology* 106:331-347.

Klump, G. M. & Gerhardt, H. C. 1992. Mechanisms and function of call-timing in male-male interactions in frogs. In: *Playback and Studies of Animal Communication* (Ed. by P. K. McGregor), pp. 153-174. New York: Plenum.

Neelon, D. P. and G. Höbel. 2019. Staying ahead of the game – plasticity in chorusing behavior allows males to remain attractive in different social environments. *Behav. Ecol. Sociobiol.* 73:124 <https://doi.org/10.1007/s00265-019-2737-1>

Schwartz, J. J. 1993. Male calling behavior, female discrimination and acoustic interference in the Neotropical treefrog *Hyla microcephala* under realistic acoustic conditions. *Behav. Ecol. Sociobiol.* 32: 401-414.

Schwartz, J. J., Buchanan, B. and H. C. Gerhardt. 2002. Acoustic interactions among male gray treefrogs (*Hyla versicolor*) in a chorus setting. *Behav. Ecol. Sociobiol.* 53:9-19.

Vélez A, Schwartz JJ, and Bee MA (2013) Anuran acoustic signal perception in noisy environments. In: Brumm H (ed) Animal Communication and Noise. Springer: New York, pp 133-185.

Decision letter (RSOS-191693.R0)

02-Dec-2019

Dear Dr Aihara,

The editors assigned to your paper ("Interaction Mechanisms Quantified from Dynamical Features of Frog Choruses") have now received comments from reviewers. We would like you to revise your paper in accordance with the referee and Associate Editor suggestions which can be found below (not including confidential reports to the Editor). Please note this decision does not guarantee eventual acceptance.

Please submit a copy of your revised paper before 25-Dec-2019. Please note that the revision deadline will expire at 00.00am on this date. If we do not hear from you within this time then it will be assumed that the paper has been withdrawn. In exceptional circumstances, extensions may be possible if agreed with the Editorial Office in advance. We do not allow multiple rounds of revision so we urge you to make every effort to fully address all of the comments at this stage. If deemed necessary by the Editors, your manuscript will be sent back to one or more of the original reviewers for assessment. If the original reviewers are not available, we may invite new reviewers.

- Data accessibility

It is a condition of publication that all supporting data are made available either as supplementary information or preferably in a suitable permanent repository. The data accessibility section should state where the article's supporting data can be accessed. This section should also include details, where possible of where to access other relevant research materials

such as statistical tools, protocols, software etc can be accessed. If the data have been deposited in an external repository this section should list the database, accession number and link to the DOI for all data from the article that have been made publicly available. Data sets that have been deposited in an external repository and have a DOI should also be appropriately cited in the manuscript and included in the reference list.

If you wish to submit your supporting data or code to Dryad (<http://datadryad.org/>), or modify your current submission to dryad, please use the following link:
<http://datadryad.org/submit?journalID=RSOS&manu=RSOS-191693>

- **Competing interests**

- **Authors' contributions**

- **Acknowledgements**

- **Funding statement**

Best regards,

on behalf of Professor Wen-Xu Wang (Associate Editor) and Kevin Padian (Subject Editor)
openscience@royalsociety.org

Reviewers' Comments to Author:

Reviewer: 1

Comments to the Author(s)

The authors investigate the interaction between acoustic signals of Japanese tree frogs. They adopt a phase-coupling model and estimate the coupling function from data. The results are interesting and relevant, and certainly, deserve publication in *RSOS*. However, an additional discussion of the approach would be appropriate. Indeed, the authors infer the phase dynamics model, while their observations are not the phases, but rather singular events (which can be described, in stochastic process theory terms, as a point process). The phase is reconstructed as piecewise linear function between the events, but this may potentially be a source of errors. A similar problem appears for reconstruction of interaction of neurons, if only the spikes are available. Recently, approaches for inferring phase dynamics from observations of spikes have been suggested [*Physical Review E*, 96, 012209, 2017], in these approaches, one, however, assumes coupling in a Winfree form, not in a Kuramoto-Daido form. I think, discussion of these issues would be appropriate.

Reviewer: 2

Comments to the Author(s)

The paper by Ota et al. addresses details of the calling dynamics within choruses of the Japanese treefrog, *Hyla japonica*. As has been the case with earlier publications from this group of researchers and their colleagues, the analysis employs sophisticated mathematics used to study dynamical systems. Of particular interest and importance is their result that the male treefrogs engage in selective attention within groups. That is, the call timing of individuals responds mostly to closer (and presumably) louder individuals and to a lesser degree to males further away. This finding is consistent with those from a handful of other studies of frogs and chorusing insects but is nevertheless valuable given the small number of such studies and the nature of the analytic approach of Ota et al. – which relies on an understanding of dynamical systems beyond that, I suspect, of most students of anuran communication. This ignorant group, unfortunately, includes me. Accordingly, although I have done my best to follow the math, I cannot judge that aspect of the paper.

The writing, although generally not bad given that the authors are Japanese, required some work on my part to improve the English. In particular, there were many statements that I found unclear and others which could benefit from less wordy prose. I also feel that the paper would be improved significantly if there was more explanation of some of the mathematics.

The findings are significant, and it would be nice for readers who are not mathematicians, engineers or physicists to be able to understand more of what particular aspects of the analysis mean and accomplish (and why they do so). Ota et al. don't neglect this entirely but additional help would be quite valuable to many in their audience. As I suggest below, the authors could effectively deal with the terminology issue by including in their paper (perhaps in an Appendix) a glossary. This could be in the form of a table or enclosed "box" of definitions also indicating how the items specifically relate to their system of vocally interacting frogs. I'll be frank. Reading the publications of Aihara and colleagues has sometimes been frustrating because the mathematics has often not been accompanied by sufficient explanation. The work is superb but it is a struggle to understand much of the content of their papers.

The spatial arrangement of males used in this study was quite simple (3 evenly spaced males in a line). Therefore, I also strongly recommend that the authors discuss limitations of their study and expectations under more realistic (i.e. natural) geometries. If data are available from some of their other studies or those published by other researchers, in their discussion section they should incorporate relevant details.

Note below that I have mentioned particular citations when I feel that additional or more appropriate research needs to be credited. A list of the references is provided at the end of my specific comments.

Specific Comments

Lines 14-26. Modify the abstract to read as follows.

“We employed empirical data from multiple recordings and a phase oscillator model to describe the deterministic and stochastic features of frog choruses in which male Japanese tree frogs attempt to avoid call overlap. The mathematical model with a general interaction term is identified using a Bayesian approach and it qualitatively reproduces the stationary and dynamical features of the empirical data. In addition, we quantify the magnitude of attention paid among the male frogs from the identified model, and then analyze the relationship between attention and behavioral parameters using a statistical approach. Our analysis demonstrates a negative correlation between attention and inter-frog distance and also suggests a behavioral strategy in which male frogs selectively attend to a less attractive male frog in order to more effectively advertise their superior relative attractiveness to females.”

Line 28. Change “form of a swarm” to “aggregations”. I have never seen or heard the term “swarm” applied to a group of birds or frogs. It is more applicable to flying insects.

Line 29 - 30. Change “flexible flock” to “flexible school or flock”. Change “To maintain the flock, they need to synchronize their velocity and direction with each other” to “To maintain the group, members need to synchronize their velocity and direction”.

Line 31. Change “also mate identification” to “also mate identification and attraction”.

Line 32. Change “acoustic signals with each other” to “acoustic signals”.

Line 34. Replace “a swarm” with “the aggregation”.

Lines 34-35. Alternation may also facilitate quality assessment by receivers in addition to be an indicator of quality. Accordingly, I recommend that you change the sentence to read as follows. “Thus, synchronization and alternation are common in animal choruses, and can both facilitate assessment of signalers and indicate their quality”.

Line 37. Change “target in a swarm” to “target in an aggregation”.

Line 40. Change “during a prey capture” to during prey capture”.

Line 41. Change “roles of the selective attention” to “roles of selective attention”. You cite two reviews [4,5]. Here or elsewhere you should mention papers that deal specifically with selective attention in frogs such as Brush and Narins (1989), Schwartz (1993), Greenfield and Rand (2000), Neelon and Hoebel (2019). Note that Schwartz et al. (2002) found that gray treefrogs (*Hyla versicolor*) do not selectively avoid overlap with their nearest neighbors.

Line 42. Change “their interaction mechanisms and compare it with the behavior of an individual animal” to “interaction mechanisms and evaluate these with respect to the behavior of an isolated animal”.

Lines 43-45. I’m not sure that here and elsewhere the word “identifying” is the best one to

describe what you did. Perhaps you should say “estimating the parameters of a phase oscillator model” to be more consistent with what you state in line 82. I’d change the sentence to read: “This study aims to quantify interaction mechanisms in choruses of male Japanese tree frogs (*Hyla japonica*) by estimating the parameters of a phase oscillator model [12] from empirical data.”

Lines 45. Change “and breed mainly at a paddy field from April to July [13]” to “and often breed in paddy fields from April to July [13]”.

Line 46. Delete the word “successive”.

Lines 47-48. Change from “demonstrate that they avoid call overlaps with each other in the forms of” to “demonstrated that they avoid overlapping calls through”.

Line 50. Here and elsewhere in your manuscript use the past tense when describing what has already been done. So, change “reveal” to “revealed”.

Lines 52-54. Change “Because the temporal overlap of acoustic signals generally mask the information included in each call, these alternating behavior would be important for male frogs to effectively advertise themselves towards a conspecific female [14-18]” to “Because the temporal overlap of acoustic signals can mask or degrade the information included in each call (Vélez et al., 2013), these alternating behaviors [14-18] can help male frogs more effectively advertise themselves to conspecific females [e.g. Schwartz, 1987].”

Are results of female choice tests available for *H. japonica*? If so, do they discriminate in favor of non-overlapping calls? Please include appropriate citations if these experiments have been conducted. You should also be aware that females of not all species prefer non-overlapping calls (see results with *H. crucifer* in Schwartz, 1987).

Line 56. Change “Japanese tree frogs from our empirical data according to a Bayesian approach” to “Japanese tree frogs based on our empirical data using a Bayesian approach”.

Line 58. Change “parameters is analyzed by a statistical approach” to “is statistically analyzed.”

Line 61. Change “choruses as the form of a” to “choruses within a”.

Lines 67-68. Change “along a straight line at the interval of 50 cm in a stationary room” to “along a straight line at intervals of 50 cm”.

Note I have no idea what you mean by “a stationary room”. Rooms do not normally move.

Lines 68-69. In your methods section, you should give the model of your microphones and recorder.

Line 70. Change “of respective frogs” to “of individual frogs”.

Line 84. So, do the results presented in figures 3-6 come from just a single data set? If so, you should make this clearer.

Lines 86-87. Change “perform numerical simulation by using the identified model and compare it with empirical data” to “performed a numerical simulation by using the identified model and compared it with empirical data” .

Line 99. Change “further analyze” to “further analyzed”.

Line 101. You need to explain what you mean by “critical states” in contrast to “equilibrium states”. I believe you are referring to your finding that a small perturbation (contributed by “added noise”) can result in an abrupt change in the form of synchronization when there are critical states present. As I mentioned above, perhaps you could most effectively deal with the terminology issue by including in your paper (perhaps in an Appendix) a table of definitions and how they specifically relate to your system of vocally interacting frogs.

Line 103. Change “this property about critical states” to “this property of critical states”. Change “origin” to “basis”.

Based on my understanding of “critical states” this argument (lines 101-103) seems circular.

Line 105. Change “we quantify” to we quantified”.

Line 106. After some checking online, I’m assuming that you are referring to the stationary distribution discussed by Kuramoto in his model of oscillators. But you really need to explain “stationary distribution” and why it is relevant. This could go in the table that I suggested you include in your paper. I realize that you say a little more about stationary distributions in section 4.5. However, more information would be very helpful so that readers can understand why the presence and form of a peak in the distribution indicates the nature of the timing between a pair of frogs and selective attention.

Change “difference is” to “difference was”.

Line 109. Change “Then, we calculate” to “Then we calculated”.

Line 116. “we analyze the relationship between the attention” to “we analyzed the relationship between attention”.

Lines 119-120. Change “(i.e., an inter-frog distance, the probability for taking a chorus leader (leader probability), and an inter-call interval)” to “(i.e., inter-frog distance, the probability of being a chorus leader (leader probability), and inter-call interval)”.

Above, I hope I have interpreted what you meant by “taking” correctly.

Line 124. Change “on the attention” to “on attention”.

Line 125. Change “the leader probability”, “the inter-call interval” and “the attention” to “leader probability”, “inter-call interval” and “attention”.

Line 138. Change “to their neighbor” to their “closer neighbor”. Given the small distance between males in your experiment, both males could be said to be neighbors. Also from the data in this paper, it is not clear what would happen when there are more than 3 males calling in a line. The Aihara et al. (2014) paper in Scientific Reports discusses this situation (at least when more males are calling) with data from your lab’s Firefly system. In your introduction, you mention the two-cluster synchronization observed in that study. However, those data were obtained from males positioned linearly along the edge of a paddy field. I’m curious as to whether you have data from any field sites (or lab tests) with males scattered in a nonlinear pattern – as would likely occur in many natural calling sites.

Lines 141-142. Again, it is important that you briefly discuss the issue of non-linear and non-even spacing intervals between males. Such spatial distributions can lead to more complex patterns of call timing between chorusing males. Schwartz (1993), for example, found that some males of *H. microcephala* (located in more interior positions) would adjust note timing in response to more than just the notes of their loudest neighbor.

Line 145. I would change “leads their chorus” to “leads another”. The studies referenced (27 and 28) used or referred to results of 2-speaker tests with females. It is not certain what would happen when more males are calling. Also, reference 28 was a study of males. A more appropriate reference would be (Greenfield and Roizen 1993).

Lines 147-148. You should check out the recent paper by Neelon and Hoebel (2019) and mention their finding that males selectively attend to the calls of more attractive males.

Line 150. Change “towards” to “to”.

Lines 152-153. Change “Thus, our analysis is likely to show the important features in frog choruses, which is valid in the contexts of the acoustic communication of male frogs as well as the strategy for mating” to “Thus, our analysis is likely to show important features in frog choruses relevant to acoustic communication of frogs and mating”.

Line 155. Change “but very close” to “but are very close”.

Line 159. Change “because of the following reasons:” to “because:”.

Line 161. Change “relying on various signals (e.g., sounds” to “relying on signals such as sounds”.

Remember to delete the closed parenthesis “)” on the next line.

Line 162. Delete “so as”.

Line 170-171. Change “inter-signal interval) when studying selective attention in animal communication that shows complicated dynamics” to “inter-signal intervals [Klump and Gerhardt, 1992] when studying selective attention when animal communication shows complicated dynamics”.

As I have indicated above, you should probably cite a reference here such as Klump and Gerhardt (1992).

Line 176. In “paddy field of Kyoto University” do you mean “at Kyoto University” or “belonging to Kyoto University” and located elsewhere?

Lines 178-179. Change “line at the interval” to “line at intervals”.

As I indicated above, I’m not sure what you mean by “a stationary room”. Most rooms are stationary. Do you mean that the positions of the caged frogs were stationary?

Line 180. Change “by three” to “using three”. You should provide information on the type (manufacturers, models) of recording equipment.

Line 183-184. Change “44 times in total” to “44 times”. Change “trials of the experiment, some of three” to “trials, some of the three”.

Line 185. Change “due to” to “using”.

Line 186-187. Change “on a sample size of the empirical data, we choose” to “on a sample size of the empirical data, we chose”.

Some clarification is also needed here. By “all the three frogs” I think you mean that the total number of calls given by the three frogs (calls of frog 1 + calls of frog 2 + calls of frog 3) was required to exceed 1400. Is this the case? Or, was each male required to give 1400 calls?

Begin a new sentence on line 187 with “The four datasets were”.

Line 188-190. Change “May in 2009, respectively. Then, the audio data of the four datasets were analyzed by the method of independent component analysis, and were separated into call signals of respective frogs [15]” to “May in 2009. The audio data of each of the four datasets were separated into call signals of individual frogs using independent component analysis [15].”

Line 191. Based on what you state in lines 200-201, I believe that you mean “three chorusing bouts from 143 chorusing bouts” rather than “three choruses from 143 choruses”. If so, make the necessary change.

Line 192. Delete the word “respective”.

Line 195. Change “guideline” to “guidelines”.

Line 199. Change “phase in the range” to “phase to the range”.

Line 201. Change “calls almost at the interval” to “calls nearly every”.

Line 203. If this is what you mean, make the following change. Substitute “incompatible with” for “out of the scope of”.

Line 205. Change “and then reset to be 0” to “and then is reset to 0”.

Line 214. Change “the term of” to “the term for”.

Line 223. Change “material for detail” to “material for details”.

Lines 241-242. Change “(1) an inter-frog distance, (2) an inter-call interval, and (3) leader-follower relation-ship. An inter-frog distance” to “(1) the inter-frog distance, (2) the inter-call interval, and (3) the leader-follower relation-ship. The inter-frog distance”

I’m curious. You always used a male-male separation of 50 cm. Have you varied the spacing so that different pairs of males are separated by different distances?

Line 247. Change “the same chorus” to “the same chorusing bout”.

Line 250. What you did is a little unclear to me. In any event, I think the following wording is better. If you agree, make the following change.

Substitute “of making each frog the chorus leader” for “for taking a chorus reader for each frog”. Note also that you misspelled “leader” as “reader”.

For the next paragraph, the continuous line numbering was absent. So, I will refer to the page-specific line numbers.

Page 13, line 20. Change “areas so as to” to “areas to”.

Page 13, line 26. Do you mean “and fixed factors” rather than “of fixed factors”? That is, do you mean that the variables you list listed are ‘fixed factors’ in your model?

Page 13, line 29. Should this be “variables and random factors” rather than “variables of random factors”?

Change “difficult to be quantified” to “difficult to quantify”.

Lines 251-252. You state “To reproduce this feature”. I’m not certain to what feature you are referring.

Line 257. Change “coefficient is less than” to “coefficient was less than”.

Line 311. For reference 18, change “begins wells” to “begins Wells”. Wells refers to the last name of Kentwood Wells and so should be capitalized.

Line 326. Change “bayesian” to “Bayesian”.

Figure Captions

For Figure 1C, males 1 and 3 do overlap. So you need to rephrase this.

Page 18. Line 53 (Caption of Figure 2). Change “dynamics according to” to “dynamics using a”.

Page 20. Line 29. Change “behavioral experiment of actual frogs” to “the behavioral experiment with actual frogs”.

Page 22. Lines 33-38. You may need to reword this figure caption based on changes you may need to make that I mention above for Page 13, line 26.

References that you should cite which are not already in your references section.

Greenfield, M. D. & Roizen, I. 1993 Katydid synchronous chorusing is an evolutionarily stable outcome of female choice. *Nature* 364, 618-620.

Greenfield MD, Rand AS (2000) Frogs have rules: Selective attention algorithms regulate chorusing in *Physalaemus pustulosus* (Leptodactylidae). *Ethology* 106:331-347.

Klump, G. M. & Gerhardt, H. C. 1992. Mechanisms and function of call-timing in male-male interactions in frogs. In: *Playback and Studies of Animal Communication* (Ed. by P. K. McGregor), pp. 153-174. New York: Plenum.

Neelon, D. P. and G. Höbel. 2019. Staying ahead of the game – plasticity in chorusing behavior allows males to remain attractive in different social environments. *Behav. Ecol. Sociobiol.* 73:124 <https://doi.org/10.1007/s00265-019-2737-1>

Schwartz, J. J. 1993. Male calling behavior, female discrimination and acoustic interference in the Neotropical treefrog *Hyla microcephala* under realistic acoustic conditions. *Behav. Ecol. Sociobiol.* 32: 401-414.

Schwartz, J. J., Buchanan, B. and H. C. Gerhardt. 2002. Acoustic interactions among male gray treefrogs (*Hyla versicolor*) in a chorus setting. *Behav. Ecol. Sociobiol.* 53:9-19.

Vélez A, Schwartz JJ, and Bee MA (2013) Anuran acoustic signal perception in noisy environments. In: Brumm H (ed) *Animal Communication and Noise*. Springer: New York, pp 133-185.

Author's Response to Decision Letter for (RSOS-191693.R0)

See Appendix A.

RSOS-191693.R1 (Revision)

Review form: Reviewer 1

Is the manuscript scientifically sound in its present form?

Yes

Are the interpretations and conclusions justified by the results?

Yes

Is the language acceptable?

Yes

Do you have any ethical concerns with this paper?

No

Have you any concerns about statistical analyses in this paper?

No

Recommendation?

Accept as is

Comments to the Author(s)

The authors satisfactory responded to my comment. I recommend the manuscript for publication.

Review form: Reviewer 2

Is the manuscript scientifically sound in its present form?

Yes

Are the interpretations and conclusions justified by the results?

Yes

Is the language acceptable?

Yes

Do you have any ethical concerns with this paper?

No

Have you any concerns about statistical analyses in this paper?

No

Recommendation?

Accept with minor revision (please list in comments)

Comments to the Author(s)

Ota et al. have done an excellent job revising their manuscript according to the comments I made in my review as well as those made by the other reviewer. However, there are a small number of language-related items to which I want to draw the attention of the authors.

Line 27: Delete the word "also".

Lines 29-20. Change “they are likely to effectively communicate and facilitate assessment of signalers within the aggregation” to “it may facilitate effective communication and assessment of signalers within the aggregation”.

Line 35” Change “when forming school” to “when forming a school”.

Line 106: Change “produce calls at specific interval” to “produce calls at a specific interval”.

Lines 106-107: Change “produce calls at unspecific interval” to “produce calls at an unspecific interval”.

Line 142: Change the title of this section from “Discussions” to “Discussion”. I failed to notice this error when I first read the manuscript.

Lines 176-178. Change “Given that a chorus leader in our definition can also avoid his calls to be masked by calls of followers at the beginning of a chorusing bout (see Figure S3 in Supplementary Information), we speculate that our result on”

to

“Given that, based on our definition of a chorus leader, a leading male can also avoid masking of his calls by those of followers at the beginning of a chorusing bout (see Figure S3 in Supplementary Information), we speculate that our result on”

Lines 181-182. Change “experiments using female *H. japonica* is required because acoustic preference of leader-follower relationship in our definition was not examined yet”

to

“experiments using female *H. japonica* are required because an acoustic preference for chorus leaders (our definition) has not yet been examined”.

Line 191: Change “understandings” to “understanding”. Change “we restricted spatial” to “we restricted the spatial”.

Lines 190-204. This section is overly wordy and repetitive. There are also some language problems. Please change it to read as follows.

“The application of our methodology to a variety of empirical data would contribute to further understanding of selective attention in frog choruses. In our laboratory experiment, we used a linear arrangement of subjects because male Japanese tree frogs are often positioned along edges of a paddy field [16]. Therefore, this was a reasonable approximation of the actual spatial distribution of the male frogs at a field site. However, unlike the evenly-spaced males in our laboratory experiment, inter-frog distance can vary among linearly arranged males at paddy fields. With respect to this point, even with a non-even distribution of Japanese treefrogs we observed that each pair of nearest neighbors tends to alternate their calls in the field [17].”
Nevertheless, because the spatial distribution of male frogs in two and three-dimensions can vary among species and chorusing venues [4], additional investigation of call timing among males under a variety of distribution patterns is certainly warranted. Given that the distance among male frogs profoundly affects the loudness of calls that other frogs recognize, the magnitude of attention is likely affected by the spatial distribution of male frogs. Related to this point...”

Line 223. Change “On the other hand, technical issue of our methodology needs to be further examined.” to “On the other hand, a technical aspect of our methodology needs to be addressed.”

Line 229. Change “Note that the similar” to “Note that a similar”.

Line 232. Change “that noise term” to “that the noise term”. Change “incorporate noisy” to “incorporate the noise”.

Line 290. Change “contain noisy component” to “contain a noise component”.

Supplementary Material

Table S1. In the box for “Noise” and “Relevance to frog choruses”, change “This term represent” to “This term represents”.

Table S2. In the box for “Kullback-Leibler divergence” and “Relevance to frog choruses”, change “at unspecific interval” to “at an unspecific interval”.

Decision letter (RSOS-191693.R1)

06-Feb-2020

Dear Dr Aihara:

On behalf of the Editors, I am pleased to inform you that your Manuscript RSOS-191693.R1 entitled "Interaction Mechanisms Quantified from Dynamical Features of Frog Choruses" has been accepted for publication in Royal Society Open Science subject to minor revision in accordance with the referee suggestions. Please find the referees' comments at the end of this email.

The reviewers and Subject Editor have recommended publication, but also suggest some minor revisions to your manuscript. Therefore, I invite you to respond to the comments and revise your manuscript.

- Ethics statement

- Data accessibility

If you wish to submit your supporting data or code to Dryad (<http://datadryad.org/>), or modify your current submission to dryad, please use the following link:
<http://datadryad.org/submit?journalID=RSOS&manu=RSOS-191693.R1>

- **Competing interests**

- **Authors' contributions**

- **Acknowledgements**

- **Funding statement**

Because the schedule for publication is very tight, it is a condition of publication that you submit the revised version of your manuscript before 15-Feb-2020. Please note that the revision deadline will expire at 00.00am on this date. If you do not think you will be able to meet this date please let me know immediately.

1) A text file of the manuscript (tex, txt, rtf, docx or doc), references, tables (including captions) and figure captions. Do not upload a PDF as your "Main Document".

- 2) A separate electronic file of each figure (EPS or print-quality PDF preferred (either format should be produced directly from original creation package), or original software format)
- 3) Included a 100 word media summary of your paper when requested at submission. Please ensure you have entered correct contact details (email, institution and telephone) in your user account
- 4) Included the raw data to support the claims made in your paper. You can either include your data as electronic supplementary material or upload to a repository and include the relevant doi within your manuscript
- 5) All supplementary materials accompanying an accepted article will be treated as in their final form. Note that the Royal Society will neither edit nor typeset supplementary material and it will be hosted as provided. Please ensure that the supplementary material includes the paper details where possible (authors, article title, journal name).

on behalf of Professor Wen-Xu Wang (Associate Editor) and Kevin Padian (Subject Editor)
openscience@royalsociety.org

Reviewer comments to Author:
Reviewer: 1

Comments to the Author(s)
The authors satisfactory responded to my comment. I recommend the manuscript for publication.

Reviewer: 2

Comments to the Author(s)
Ota et al. have done an excellent job revising their manuscript according to the comments I made in my review as well as those made by the other reviewer. However, there are a small number of language-related items to which I want to draw the attention of the authors.

Line 27: Delete the word "also".

Lines 29-20. Change "they are likely to effectively communicate and facilitate assessment of signalers within the aggregation" to "it may facilitate effective communication and assessment of signalers within the aggregation".

Line 35" Change "when forming school" to "when forming a school".

Line 106: Change “produce calls at specific interval” to “produce calls at a specific interval”.

Lines 106-107: Change “produce calls at unspecific interval” to “produce calls at an unspecific interval”.

Line 142: Change the title of this section from “Discussions” to “Discussion”. I failed to notice this error when I first read the manuscript.

Lines 176-178. Change “Given that a chorus leader in our definition can also avoid his calls to be masked by calls of followers at the beginning of a chorusing bout (see Figure S3 in Supplementary Information), we speculate that our result on”

to

“Given that, based on our definition of a chorus leader, a leading male can also avoid masking of his calls by those of followers at the beginning of a chorusing bout (see Figure S3 in Supplementary Information), we speculate that our result on”

Lines 181-182. Change “experiments using female *H. japonica* is required because acoustic preference of leader-follower relationship in our definition was not examined yet”

to

“experiments using female *H. japonica* are required because an acoustic preference for chorus leaders (our definition) has not yet been examined”.

Line 191: Change “understandings” to “understanding”. Change “we restricted spatial” to “we restricted the spatial”.

Lines 190-204. This section is overly wordy and repetitive. There are also some language problems. Please change it to read as follows.

“The application of our methodology to a variety of empirical data would contribute to further understanding of selective attention in frog choruses. In our laboratory experiment, we used a linear arrangement of subjects because male Japanese tree frogs are often positioned along edges of a paddy field [16]. Therefore, this was a reasonable approximation of the actual spatial distribution of the male frogs at a field site. However, unlike the evenly-spaced males in our laboratory experiment, inter-frog distance can vary among linearly arranged males at paddy fields. With respect to this point, even with a non-even distribution of Japanese treefrogs we observed that each pair of nearest neighbors tends to alternate their calls in the field [17].”
Nevertheless, because the spatial distribution of male frogs in two and three-dimensions can vary among species and chorusing venues [4], additional investigation of call timing among males under a variety of distribution patterns is certainly warranted. Given that the distance among male frogs profoundly affects the loudness of calls that other frogs recognize, the magnitude of attention is likely affected by the spatial distribution of male frogs. Related to this point...”

Line 223. Change “On the other hand, technical issue of our methodology needs to be further examined.” to “On the other hand, a technical aspect of our methodology needs to be addressed.”

Line 229. Change “Note that the similar” to “Note that a similar”.

Line 232. Change “that noise term” to “that the noise term”. Change “incorporate noisy” to “incorporate the noise”.

Line 290. Change “contain noisy component” to “contain a noise component”.

Supplementary Material

Table S1. In the box for “Noise” and “Relevance to frog choruses”, change “This term represent” to “This term represents”.

Table S2. In the box for “Kullback-Leibler divergence” and “Relevance to frog choruses”, change “at unspecific interval” to “at an unspecific interval”.

Author's Response to Decision Letter for (RSOS-191693.R1)

See Appendix B.

Decision letter (RSOS-191693.R2)

27-Feb-2020

Dear Dr Aihara,

It is a pleasure to accept your manuscript entitled "Interaction Mechanisms Quantified from Dynamical Features of Frog Choruses" in its current form for publication in Royal Society Open Science. The comments of the reviewer(s) who reviewed your manuscript are included at the foot of this letter.

Kind regards,
Anita Kristiansen
Editorial Coordinator
Royal Society Open Science
openscience@royalsociety.org

on behalf of Professor Wen-Xu Wang (Associate Editor) and Kevin Padian (Subject Editor)
openscience@royalsociety.org

Appendix A

Dear Prof. Wen-Xu Wang and Prof. Kevin Padian,

We are pleased to send you the revised version of our manuscript. We have made most of the changes and corrections suggested by the reviewers. Given their feedback we also added some clarifications. We feel that the manuscript has improved substantially thanks to their feedback.

In the response to each reviewer (below), we address each one of their concerns independently in details. In particular, we responded to their main concerns as follows:

- (i) ***Observation error caused by a piecewise linear interpolation (pointed out by Reviewer 1):*** The 1st reviewer suggested us to mention observation error caused by a piecewise linear interpolation. While we had confirmed the validity of the present method from various points of view (e.g., comparison between numerical simulation and empirical data, and the analysis on synchronized features on the identified model) in this case of frog choruses, we agree with the comment that further discussion is necessary. Subsequently, we added a new paragraph to the section of Discussion (Lines 225-235). In the new paragraph, we first mention that observation error can be caused by the linear interpolation in this study, and then explain how our method likely deals with the error within the framework of a phase oscillator model.
- (ii) ***Terminology issue (pointed by Reviewer 2):*** We agree with the comment that further explanation on terminology helps readers to understand the mathematical parts of this study. To improve this point, we added detailed explanation on important terminology with mentioning its relevance to frog choruses, and then summarized the explanation in tables that we newly added to Supplementary Information according to the comment of the 2nd reviewer. Please also confirm our responses to Comments 27, 28, and 30 of the 2nd reviewer for details.
- (iii) ***Limitation of spatial arrangement in our laboratory experiment (pointed out by Reviewer 2):*** The 2nd reviewer pointed out that we need to further discuss the variation in spatial distribution of male frogs because we restricted the spatial distribution to a linear arrangement in our laboratory experiment. We basically agree with this comment. As for this point, we had published a paper reporting that spatial distribution of male Japanese tree frogs is almost linear (namely, they are often positioned along edges of a paddy field [Aihara et al. 2014]). Therefore, the linear

arrangement that we employed in this study is a reasonable approximation of the actual distribution observed in a field site. In the revised manuscript, we added these explanations on the limitation and its reason about our spatial arrangement, and then mention the future direction of this study related to complicated spatial distribution with many frogs that can occur in natural environment especially in other frog species (Lines 190-210). Please confirm our responses to Comments 37 and 38 of the 2nd reviewer for details.

Overall, we appreciate the reviewers' comments and suggestions since they have greatly improved the quality of our study. A marked-up copy of the changes from the previous manuscript is also attached as a different file (Marked-up-copy.pdf).

We greatly appreciate your feedback and collaboration.

Yours Sincerely,

Ikkyu Aihara (for all the authors)

Our responses to the comments of Reviewer 1

General comment:

The authors investigate the interaction between acoustic signals of Japanese tree frogs. They adopt a phase-coupling model and estimate the coupling function from data. The results are interesting and relevant, and certainly, deserve publication in RSOS. However, an additional discussion of the approach would be appropriate. Indeed, the authors infer the phase dynamics model, while their observations are not the phases, but rather singular events (which can be described, in stochastic process theory terms, as a point process). The phase is reconstructed as piecewise linear function between the events, but this may potentially be a source of errors. A similar problem appears for reconstruction of interaction of neurons, if only the spikes are available. Recently, approaches for inferring phase dynamics from observations of spikes have been suggested [Physical Review E, 96, 012209, 2017], in these approaches, one, however, assumes coupling in a Winfree form, not in a Kuramoto-Daido form. I think, discussion of these issues would be appropriate.

Response:

Thank you very much for your valuable comment. As you mention, we reconstructed a phase from point process (i.e., calling events of male frogs) based on piecewise linear interpolation (Equation (1)). Because we did not directly observe the phase, we agree with your comment that the piecewise linear interpolation can be a source of observation error. To discuss this issue for details, we added a new paragraph to the section of Discussion (Lines 225-235).

In the new paragraph, we first mention observation error caused by the linear interpolation, and then introduce the relevance with other biological oscillators like spiking neurons. In addition, we explain our speculation that, in this study, noise term of a phase oscillator model would incorporate noisy component including the observation error. To further examine the validity and limitation of the proposed method, however, we need to compare the performance of our method with that of other methods. In the new paragraph, we also discuss this point with referring a relevant study that you mention [Physical Review E, 96, 012209, 2017].

Thank you very much again for your valuable comment which indeed helps us to improve the manuscript.

Our responses to the comments of Reviewer 2

General comment:

The paper by Ota et al. addresses details of the calling dynamics within choruses of the Japanese treefrog, *Hyla japonica*. As has been the case with earlier publications from this group of researchers and their colleagues, the analysis employs sophisticated mathematics used to study dynamical systems. Of particular interest and importance is their result that the male treefrogs engage in selective attention within groups. That is, the call timing of individuals responds mostly to closer (and presumably) louder individuals and to a lesser degree to males further away. This finding is consistent with those from a handful of other studies of frogs and chorusing insects but is nevertheless valuable given the small number of such studies and the nature of the analytic approach of Ota et al. – which relies on an understanding of dynamical systems beyond that, I suspect, of most students of anuran communication. This ignorant group, unfortunately, includes me. Accordingly, although I have done my best to follow the math, I cannot judge that aspect of the paper.

The writing, although generally not bad given that the authors are Japanese, required some work on my part to improve the English. In particular, there were many statements that I found unclear and others which could benefit from less wordy prose. I also feel that the paper would be improved significantly if there was more explanation of some of the mathematics.

The findings are significant, and it would be nice for readers who are not mathematicians, engineers or physicists to be able to understand more of what particular aspects of the analysis mean and accomplish (and why they do so). Ota et al. don't neglect this entirely but additional help would be quite valuable to many in their audience. As I suggest below, the authors could effectively deal with the terminology issue by including in their paper (perhaps in an Appendix) a glossary. This could be in the form of a table or enclosed "box" of definitions also indicating how the items specifically relate to their system of vocally interacting frogs. I'll be frank. Reading the publications of Aihara and colleagues has sometimes been frustrating because the mathematics has often not been accompanied by sufficient explanation. The work is superb but it is a struggle to understand much of the content of their papers.

The spatial arrangement of males used in this study was quite simple (3 evenly spaced males in a line). Therefore, I also strongly recommend that the authors discuss limitations of their study and

expectations under more realistic (i.e. natural) geometries. If data are available from some of their other studies or those published by other researchers, in their discussion section they should incorporate relevant details.

Note below that I have mentioned particular citations when I feel that additional or more appropriate research needs to be credited. A list of the references is provided at the end of my specific comments.

Response:

We appreciate very much your careful readings of the manuscript as well as valuable comments based on your wide knowledges and great perspectives on the behavior of frogs. As described below, we responded to all the specific comments of you, and added modification to our manuscript. In particular, we responded to your main concerns as follows:

- (i) ***Terminology issue:*** We agree with your comments that further explanation on terminology is necessary and it can help readers to well understand the mathematical parts of this study. To improve this point, we added detailed explanation on terminology used in this study (e.g., an equilibrium state, a critical state, and stationary distribution of a phase difference) with mentioning its relevance to frog choruses, and then summarized those explanation in tables (Tables S1-S3) that we newly added to Supplementary Information. As for the explanation of an equilibrium state and a critical state, we also added schematic figure to Supplementary Information (Figure S2 in Supplementary Information). Please confirm our responses to Comments 27, 28, and 30 for details.
- (ii) ***Limitation of spatial arrangement in our laboratory experiment:*** We also agree with your comments suggesting that we need to further discuss the variation in spatial distribution of male frogs because we restricted the spatial distribution to linear arrangement in our laboratory experiment. Subsequently, we added a new paragraph to the section of Discussion (**Lines 190-210**). In the new paragraph, we first mention the restriction of spatial arrangement that we employed in this study, explain the reason of the restriction, and then state the future direction including the application of our methodology to field data recorded in more complicated spatial distribution with more frogs. While we do not currently have field data recorded in the complicated spatial distribution such as scattering pattern at non-even inter-frog

distance unfortunately, we try to explain the future direction in details with mentioning relevance to the behavior of other frog species that you suggest in Comments 10 and 38. Please confirm our responses to Comments 37 and 38 for details.

- (iii) *English and references*: We indeed appreciate your careful readings of our manuscript and valuable comments on English and references. In this revise, we modified English of our manuscript according to your comments step by step, and then checked whole the manuscript to add the same modification to related phrases. In addition, we carefully read relevant papers that you suggested, and agree with you that we should refer them. For this resubmission, we also attached a marked-up copy of the changes from the previous manuscript "marked-up-copy.pdf" in which we highlighted the additional references with blue font.

Thank you very much again for all the comments and suggestions that greatly help us to improve the manuscript.

Specific Comments:

Comment 1:

Lines 14-26. Modify the abstract to read as follows.

“We employed empirical data from multiple recordings and a phase oscillator model to describe the deterministic and stochastic features of frog choruses in which male Japanese tree frogs attempt to avoid call overlap. The mathematical model with a general interaction term is identified using a Bayesian approach and it qualitatively reproduces the stationary and dynamical features of the empirical data. In addition, we quantify the magnitude of attention paid among the male frogs from the identified model, and then analyze the relationship between attention and behavioral parameters using a statistical approach. Our analysis demonstrates a negative correlation between attention and inter-frog distance and also suggests a behavioral strategy in which male frogs selectively attend to a less attractive male frog in order to more effectively advertise their superior relative attractiveness to females.”

Response:

Thank you very much for the valuable comment on the abstract. We modified the abstract

according to your comment (Lines 14-22). Note that this modification is also based on our response to your 39th comment.

Comment 2:

Line 28. Change “form of a swarm” to “aggregations”. I have never seen or heard the term “swarm” applied to a group of birds or frogs. It is more applicable to flying insects.

Response:

Thank you very much for the comment. It was our mistake that we used the word "swarm" to describe the aggregation of frogs. We now agree with your comment pointing out that the word "aggregations" is more appropriate. In the revised manuscript, we replaced the word "form of a swarm" with "aggregations" in the introduction (Line 24), and added the same modification throughout the manuscript when describing the aggregation of frogs.

Comment 3:

Line 29 - 30. Change “flexible flock” to “flexible school or flock”. Change “To maintain the flock, they need to synchronize their velocity and direction with each other” to “To maintain the group, members need to synchronize their velocity and direction”.

Response:

We changed the phrases according to your comment (Lines 25 and 26).

Comment 4:

Line 31. Change “also mate identification” to “also mate identification and attraction”.

Response:

Thank you for the comment. We changed the phrase (Line 27).

Comment 5:

Line 32. Change “acoustic signals with each other” to “acoustic signals”.

Response:

We changed the phrase according to your comment (Line 28).

Comment 6:

Line 34. Replace “a swarm” with “the aggregation”.

Response:

We changed the word (Line 30).

Comment 7:

Lines 34-35. Alternation may also facilitate quality assessment by receivers in addition to be an indicator of quality. Accordingly, I recommend that you change the sentence to read as follows. “Thus, synchronization and alternation are common in animal choruses, and can both facilitate assessment of signalers and indicate their quality”.

Response:

Thank you very much for your suggestion. First of all, we absolutely agree with your comment pointing out that alternation may facilitate assessment of signalers in animal choruses. On the other hand, in this paragraph, we introduce examples of synchronization (corresponding to spatial alignment in animals (i.e., formation of bird flock and fish school)) and alternation (corresponding to acoustic communication in animals). The sentence that you mention summarizes the roles of both examples, and then is not restricted on the acoustic communication in animals. Therefore, in the revised manuscript, we first added the description on the role of facilitating assessment of signalers in acoustic communication, and then modify the wording in the last sentence of this paragraph according to your comment (Lines 30-32).

Comment 8:

Line 37. Change “target in a swarm” to “target in an aggregation”.

Response:

We changed the phrase (Line 33).

Comment 9:

Line 40. Change “during a prey capture” to during prey capture”.

Response:

Thank you. We deleted the indefinite article "a" from the phrase (Line 36).

Comment 10:

Line 41. Change “roles of the selective attention” to “roles of selective attention”. You cite two reviews [4,5]. Here or elsewhere you should mention papers that deal specifically with selective attention in frogs such as Brush and Narins (1989), Schwartz (1993), Greenfield and Rand (2000), Neelon and Hoebel (2019). Note that Schwartz et al. (2002) found that gray treefrogs (*Hyla versicolor*) do not selectively avoid overlap with their nearest neighbors.

Response:

Thank you for the comment. We first changed the phrase according to your comment (Lines 37 and 38). Then, we agree with your comment suggesting that we should refer five papers [Brush and Narins 1989], [Schwartz 1993], [Greenfield and Rand 2000], [Neelon and Hoebel 2019] and [Schwartz et al., 2002] in the context of selective attention. Subsequently, we referred four of them ([Brush and Narins 1989], [Schwartz 1993], [Greenfield and Rand 2000] and [Neelon and Hoebel 2019]) in the same paragraph as related studies on selective attention in male frogs. In addition, we referred another paper ([Schwartz et al., 2002]) in the section of Discussion, and mentioned that selective attention among nearest neighbors is not always observed as in the case of male *Hyla versicolor* (Lines 207 and 208).

Comment 11:

Line 42. Change “their interaction mechanisms and compare it with the behavior of an individual animal” to “interaction mechanisms and evaluate these with respect to the behavior of an isolated animal”.

Response:

Thank you for the comment. We modified the phrase based on your comment (Lines 38 and 39). Note that we slightly changed the wording from your suggestion by replacing "the behavior of an isolated animal" with "the behavior of interacting animals". This is because here we want to mention the importance of utilizing behavioral parameters of interacting animals as we did in this study (namely, we estimated three behavioral parameters of inter-frog distance, inter-call intervals and leader-follower relationship from empirical data of interacting male frogs, and then utilized those parameters for a statistical analysis). We recognize that the word "isolated" means the situation in which only one animal exists in a space; therefore we changed the phrase "of an isolated animal" to "of interacting animals". As for this sentence, we apologize that the phrase "of an individual animal" that we used in the previous manuscript was inappropriate.

Comment 12:

Lines 43-45. I'm not sure that here and elsewhere the word "identifying" is the best one to describe what you did. Perhaps you should say "estimating the parameters of a phase oscillator model" to be more consistent with what you state in line 82. I'd change the sentence to read: "This study aims to quantify interaction mechanisms in choruses of male Japanese tree frogs (*Hyla japonica*) by estimating the parameters of a phase oscillator model [12] from empirical data."

Response:

Thank you for the comment. In the previous manuscript, we used the phrase "identifying a phase oscillator model" as the same meaning with "estimating the parameters of a phase oscillator model". We agree with your comment that the phrase "estimating the parameters of a phase oscillator model" is more appropriate. Hence, we modified the phrase according to your comment (Line 41), and then added the same modification throughout the manuscript (see a marked-up copy of the changes from the previous manuscript "marked-up-copy.pdf" for details).

Comment 13:

Lines 45. Change "and breed mainly at a paddy field from April to July [13]" to "and often breed

in paddy fields from April to July [13]”.

Response:

We changed the phrase (Lines 42 and 43).

Comment 14:

Line 46. Delete the word “successive”.

Response:

We deleted the word (Line 43).

Comment 15:

Lines 47-48. Change from “demonstrate that they avoid call overlaps with each other in the forms of” to “demonstrated that they avoid overlapping calls through”.

Response:

We changed the phrase (Lines 44 and 45).

Comment 16:

Line 50. Here and elsewhere in your manuscript use the past tense when describing what has already been done. So, change “reveal” to “revealed”.

Response:

Thank you very much for your valuable comment. We modified the verb tense according to your comment (Line 47). In addition, we carefully checked the tense throughout the manuscript, and added necessary modifications (see a marked-up copy of the changes from the previous manuscript "marked-up-copy.pdf" for details).

Comment 17:

Lines 52-54. Change “Because the temporal overlap of acoustic signals generally mask the

information included in each call, these alternating behavior would be important for male frogs to effectively advertise themselves towards a conspecific female [14–18]” to

“Because the temporal overlap of acoustic signals can mask or degrade the information included in each call (Vélez et al., 2013), these alternating behaviors [14–18] can help male frogs more effectively advertise themselves to conspecific females [e.g. Schwartz, 1987].”

Are results of female choice tests available for *H. japonica*? If so, do they discriminate in favor of non-overlapping calls? Please include appropriate citations if these experiments have been conducted. You should also be aware that females of not all species prefer non-overlapping calls (see results with *H. crucifer* in Schwartz, 1987).

Response:

Thank you for the comment. We have carefully read the two papers [Vélez et al., 2013] and [Schwartz, 1987], and agree with your comment suggesting that we should refer them. Subsequently, we changed the sentence according to your comment with referring [Vélez et al., 2013] and [Schwartz, 1987] (Lines 48-51). Note that we specified that Schwartz reported acoustic preference on non-overlapping calls in two species *H. microcephala* and *H. versicolor*. We also referred a recent paper [Leggett et al., 2019] reporting the similar acoustic preference in female tungara frogs (*E. pustulosus*).

On the other hand, to our knowledge, acoustic preference of female *H. japonica* was not examined well unfortunately. For this revision, we carefully searched papers on *H. japonica* again, but could not find relevant one. We feel that other researchers had the similar issue. For instance, An and Waldman examined the change in calling behavior of male *H. japonica* infected by chytrid fungus [An and Waldman, *Biology Letters*, 2016]. In the paper, they introduce their speculation on the preference of female *H. japonica* on call traits (call repetition rate) of conspecific males, but they don't refer a relevant paper on *H. japonica* and restrict their discussion to general description of female frogs. Consequently, we had to also restrict our discussion to the general description of female frogs in this manuscript as in Lines 159 and 160.

Comment 18:

Line 56. Change “Japanese tree frogs from our empirical data according to a Bayesian approach” to “Japanese tree frogs based on our empirical data using a Bayesian approach”.

Response:

We changed the phrase (Line 53).

Comment 19:

Line 58. Change “parameters is analyzed by a statistical approach” to “is statistically analyzed.”

Response:

We changed the phrase (Line 55).

Comment 20:

Line 61. Change “choruses as the form of a” to “choruses within a”.

Response:

We changed the phrase (Line 59).

Comment 21:

Lines 67-68. Change “along a straight line at the interval of 50 cm in a stationary room” to “along a straight line at intervals of 50 cm”.

Note I have no idea what you mean by “a stationary room”. Rooms do not normally move.

Response:

Thank you for the comment. We changed the phrase according to your comment (Lines 65 and 66). In the previous manuscript, we used the word "stationary" to emphasize that condition of a room (e.g., illuminance and temperature) was stationary. However, such a condition is also normally stationary in a room, and we consider that it is not necessary to emphasize it here. Therefore, we deleted the word "in a stationary room" from this sentence. Note that we modified this sentence also according to our response to your 66th comment.

Comment 22:

Lines 68-69. In your methods section, you should give the model of your microphones and recorder.

Response:

Thank you very much for the comment. We added the information of model for both microphones and recorder in the section of Materials and Methods (Lines 244-246).

Comment 23:

Line 70. Change “of respective frogs” to “of individual frogs”.

Response:

We changed the phrase (Line 68).

Comment 24:

Line 84. So, do the results presented in figures 3-6 come from just a single data set? If so, you should make this clearer.

Response:

Thank you very much for the valuable comment. Yes, Figures 3-6 come from a single dataset while Figure 7 summarizes the result of all the datasets. To clarify this point, we added a new sentence to the section of Results (Lines 80-83).

Comment 25:

Lines 86-87. Change “perform numerical simulation by using the identified model and compare it with empirical data” to “performed a numerical simulation by using the identified model and compared it with empirical data” .

Response:

We changed the phrase (Lines 85 and 86). We indeed appreciate your careful reading on the manuscript.

Comment 26:

Line 99. Change “further analyze” to “further analyzed”.

Response:

We changed the verb tense (Line 107).

Comment 27:

Line 101. You need to explain what you mean by “critical states” in contrast to “equilibrium states”. I believe you are referring to your finding that a small perturbation (contributed by “added noise”) can result in an abrupt change in the form of synchronization when there are critical states present. As I mentioned above, perhaps you could most effectively deal with the terminology issue by including in your paper (perhaps in an Appendix) a table of definitions and how they specifically relate to your system of vocally interacting frogs.

Response:

Thank you very much. We agree with your valuable comment suggesting that a table of terminology can help readers to well understand the mathematical parts of this study. For this revision, we newly made three tables describing the terminology in a phase oscillator model (Table S1 in Supplementary Information), that in the time differential equation of a phase difference (Table S2), and that in the Fokker-Plank equation (Table S3); each of the tables includes definition and also relevance to frog choruses as you suggest.

Then, we explain the difference between an equilibrium state and a critical state in details. First, an equilibrium state ($\psi_{nm} = \psi_{nm}^*$) describes the situation in which the phase difference converges to ψ_{nm}^* and then remain around ψ_{nm}^* forever, which corresponds to the behavior that two males produce calls at specific phase difference ψ_{nm}^* quite robustly. Second, a critical state ($\psi_{nm} = \psi_{nm}^{**}$) describes the situation in which the phase difference stays around ψ_{nm}^{**} for a long time, and intermittently leaves the point, which corresponds to the behavior that two males produce calls at specific phase difference ψ_{nm}^{**} for a long time and intermittently produce calls at unspecific phase difference. Thus, the stability of a critical state is weaker than that of an equilibrium state. To explain this feature in details, we added explanation

on the stabilities of an equilibrium state and a critical state to the main manuscript (Lines 99-113), provided a new schematic figure explaining the difference between an equilibrium state and a critical state (Figure S2 in Supplementary Information), and then summarized the above explanation in a table of terminology (Table S2 in Supplementary Information).

On the other hand, an equilibrium state and a critical state can be defined independently of the existence of noise. As for this point, we apologize that the previous description "*Because the transitions among multiple states can be driven by added noise under the existence of critical states in various dynamical systems, this property about critical states is likely to be the origin of the transitions observed in the empirical data of the frog choruses.*" was confusing. The messages that we want to say here is twofold: (1) the stability of a critical state is weaker than that of an equilibrium state, and then (2) critical states are likely more sensitive to added noise. To make this point more clear, we changed the structure of the sentence (Lines 111-113). Note that we also incorporated your 28th comment (the next one) for the revision of this sentence.

Comment 28:

Line 103. Change “this property about critical states” to “this property of critical states”. Change “origin” to “basis”.

Based on my understanding of “critical states” this argument (lines 101-103) seems circular.

Response:

Thank you very much again for your valuable comment. As for this sentence, we first added modification on the sentence on the basis of our response to Comment 37, and then modified the wording according to your comment (Line 111). As you correctly interpret, a critical state is more dynamical than an equilibrium state because of its weaker stability.

Comment 29:

Line 105. Change “we quantify” to we quantified”.

Response:

We changed the verb tense (Line 114).

Comment 30:

Line 106. After some checking online, I'm assuming that you are referring to the stationary distribution discussed by Kuramoto in his model of oscillators. But you really need to explain "stationary distribution" and why it is relevant. This could go in the table that I suggested you include in your paper. I realize that you say a little more about stationary distributions in section 4.5. However, more information would be very helpful so that readers can understand why the presence and form of a peak in the distribution indicates the nature of the timing between a pair of frogs and selective attention.

Response:

Thank you very much. We agree with your comment suggesting that we need additional explanations on the stationary distribution with utilizing a table of terminology in supplementary information. In this study, from the identified phase oscillator model, we obtained the Fokker-Plank equation **that governs the time evolution of the transient distribution of a phase difference (i.e., $f(\psi_{nm}, t)$)**, and then calculated the stationary distribution of $f(\psi_{nm}, t)$ as $\hat{f}(\psi_{nm})$ by numerically solving the Fokker-Plank equation until $f(\psi_{nm}, t)$ had converged. Thus, the word "stationary" means that the distribution does not change any more in time and then it is stable, and therefore the stationary distribution $\hat{f}(\psi_{nm})$ provides **the distribution of a phase difference that is most expected to be realized by the identified model**. More specifically, if the stationary distribution has a peak around π , it is very likely that a male frog attempted to alternate calls with another male frog.

To clarify these points, we added the above explanation to the section of Results (**Lines 114-121**) and also to the section of Materials and Methods (**Lines 345-358**), and summarized them in a table of terminology in Supplementary Information (see Table S3 in Supplementary Information). Thank you very much again for your valuable comments that greatly help us to strengthen the explanation on mathematical parts of this study.

Comment 31:

Change "difference is" to "difference was".

Response:

We changed the verb tense (Line 115).

Comment 32:

Line 109. Change “Then, we calculate” to “Then we calculated”.

Response:

We changed the phrase (Line 122).

Comment 33:

Line 116. “we analyze the relationship between the attention” to “we analyzed the relationship between attention”.

Response:

We changed the phrase (Lines 128 and 129).

Comment 34:

Lines 119-120. Change “(i.e., an inter-frog distance, the probability for taking a chorus leader (leader probability), and an inter-call interval)” to “(i.e., inter-frog distance, the probability of being a chorus leader (leader probability), and inter-call interval)”.

Above, I hope I have interpreted what you meant by “taking” correctly.

Response:

Thank you very much. We changed the phrase according to your comment (Lines 130 and 131).

Comment 35:

Line 124. Change “on the attention” to “on attention”.

Response:

We changed the phrase (Line 135).

Comment 36:

Line 125. Change “the leader probability”, “the inter-call interval” and “the attention” to “leader probability”, “inter-call interval” and “attention”.

Response:

We changed the phrases according to your comment (Lines 136 and 137).

Comment 37:

Line 138. Change “to their neighbor” to their “closer neighbor”. Given the small distance between males in your experiment, both males could be said to be neighbors. Also from the data in this paper, it is not clear what would happen when there are more than 3 males calling in a line. The Aihara et al. (2014) paper in Scientific Reports discusses this situation (at least when more males are calling) with data from your lab’s Firefly system. In your introduction, you mention the two-cluster synchronization observed in that study. However, those data were obtained from males positioned linearly along the edge of a paddy field. I’m curious as to whether you have data from any field sites (or lab tests) with males scattered in a nonlinear pattern – as would likely occur in many natural calling sites.

Response:

Thank you for the valuable comment. First, we changed the phrase to "their closer neighbor" according to the comment (Line 154). Next, we added a new paragraph to the section of *Discussion* (Lines 190-210) in order to state the limitation of the spatial arrangement that we employed in this study. As for this point, we also agree with your comment suggesting the importance of the studies on complicated spatial distribution with more frogs.

As you mention, we restricted the spatial distribution of male *H. japonica* to a linear arrangement. It was because the distribution of male *H. japonica* is almost linear in a field site (i.e., males are often positioned along edges of a paddy field) as you mention with referring our previous study [Aihara et al., *Scientific Reports*, 2014], and then the linear arrangement of this

study is likely a reasonable approximation on the actual distribution of male Japanese tree frogs. In the new paragraph, we first mention these features on limitation and its reason about the spatial arrangement of male frogs. Then, we point out the importance of the studies on different spatial distribution such as scattering nonlinear pattern with many frogs while we do not currently have such a data on male Japanese tree frogs unfortunately. As for this point, we are sure that it is the most important future direction to apply our methodology to field data of multiple frog species because the spatial distribution of male frogs can vary a lot.

Comment 38:

Lines 141-142. Again, it is important that you briefly discuss the issue of non-linear and non-even spacing intervals between males. Such spatial distributions can lead to more complex patterns of call timing between chorusing males. Schwartz (1993), for example, found that some males of *H. microcephala* (located in more interior positions) would adjust note timing in response to more than just the notes of their loudest neighbor.

Response:

Thank you for the comment. As explained in our response to your 37th comment (the previous one), we added a new paragraph and discussed the effect of complexity in spatial distribution (Lines 190-210). As for this point, we have carefully read the paper [Schwartz 1993], and agree with your comment that we should refer them. In the new paragraph, we referred the papers [Schwartz 1993] and [Schwartz et al. 2002] as relevant studies and mentioned the existence of further complexity in selective attention among species (Note that, in Comment 10, you mentioned [Schwartz et al. 2002] as an example of no selective attention in male frogs). Thank you very much again for informing the important relevant studies that we should refer.

Comment 39:

Line 145. I would change “leads their chorus” to “leads another”. The studies referenced (27 and 28) used or referred to results of 2-speaker tests with females. It is not certain what would happen when more males are calling. Also, reference 28 was a study of males. A more appropriate reference would be (Greenfield and Roizen 1993).

Response:

Thank you very much for your comment. I carefully read the paper [Greenfield and Roizen 1993] that you suggest, and agree with your comment. In the revised manuscript, we refer the paper instead of Reference 28 in the previous manuscript, as a relevant study on the acoustic preference of female frogs to a leading call (Line 176).

On the other hand, a leader-follower relationship assumed in this manuscript is related to but is different from traditional definition. We apologize that we could not recognize this difference when we had initially submitted the paper. In this study, we define a chorus leader as a male that started calling earlier than other males within the same chorusing bout as explained in the manuscript (Lines 369-372). In contrast, a chorus leader is traditionally defined on the basis of relationship between adjacent calls of male frogs; for instance, if Frog 1 partially overlaps his call with Frog 2 but produces the call slightly earlier than him, Frog 1 is defined as a chorus leader. To explain this difference on the definition of leader-follower relationship, we added a new paragraph to the section of Discussion (Lines 166-182); we then added supplementary figure to clarify the definition of leader-follower relationship in this study (Figure S3 in Supplementary Information). Given that a chorus leader can also avoid his calls to be masked by calls of followers at the beginning of a chorusing bout in our definition, we speculate that a chorus leader in our definition is also more attractive to female frogs than followers. However, further studies on the acoustic preference to a chorus leader in our definition are required because the preference was not examined yet. In the added paragraph, we also mention this speculation (Lines 176-180) as well as the limitation of this study (Lines 180-182). Associated with this update, in the abstract we avoided mentioning the acoustic preference of female frogs in the context of leader-follower relationship (Line 21).

Comment 40:

Lines 147-148. You should check out the recent paper by Neelon and Hoebel (2019) and mention their finding that males selectively attend to the calls of more attractive males.

Response:

Thank you very much for informing an important paper. We carefully read the paper, and confirmed that they worked on a relevant system, namely on the attention of male frogs on two sound sources. While their finding is inconsistent with our manuscript, we agree with your

comment that we should mention it. In the revised manuscript, we added sentences to mention that Neelon and Hoebel also worked on attention of male frogs by playback experiments and reported an inconsistent result with our manuscript (Lines 184-186). On the other hand, because Neelon and Hoebel studied the behavior of different frog species (*H. cinerea*) and utilize other call trait (i.e., sound frequency) as the indicator of attractiveness, we feel that comprehensive studies on various frog species are required to further examine the attention of chorusing males. We also mention this point in the same paragraph(Lines 186-189).

Comment 41:

Line 150. Change “towards” to “to”.

Response:

We changed the word (Line 164).

Comment 42:

Lines 152-153. Change “Thus, our analysis is likely to show the important features in frog choruses, which is valid in the contexts of the acoustic communication of male frogs as well as the strategy for mating” to “Thus, our analysis is likely to show important features in frog choruses relevant to acoustic communication of frogs and mating”.

Response:

We modified the sentence based on your comment (Lines 183 and 184). Note that we slightly simplified the phrase from your suggestion, by shortening "acoustic communication of frogs and mating" to "acoustic communication and mating".

Comment 43:

Line 155. Change “but very close” to “but are very close”.

Response:

Thank you very much. We changed the phrase according to your comment (Line 138). Note that,

in the revised manuscript, we moved the description on the confidence interval of Bayesian estimation from Discussion to Results because it is directly related to the result of the Bayesian estimation rather than the discussion (Lines 137-141).

Comment 44:

Line 159. Change “because of the following reasons:” to “because:”.

Response:

We changed the phrase (Line 212).

Comment 45:

Line 161. Change “relying on various signals (e.g., sounds” to “relying on signals such as sounds”.

Remember to delete the closed parenthesis “)” on the next line.

Response:

Thank you. We changed the phrase according to your comment (Lines 213 and 214).

Comment 46:

Line 162. Delete “so as”.

Response:

We deleted the words (Line 215).

Comment 47:

Line 170-171. Change “inter-signal interval) when studying selective attention in animal communication that shows complicated dynamics” to “inter-signal intervals [Klump and Gerhardt, 1992]) when studying selective attention when animal communication shows complicated dynamics”.

As I have indicated above, you should probably cite a reference here such as Klump and Gerhardt (1992).

Response:

Thank you for the comment. We modified the wording on the basis of your comment (Lines 221-224). Note that, in order to simplify the structure of the sentence, we slightly changed the phrase from your suggestion, as "We believe that this property of a dynamical model is advantageous **for the analysis on selective attention** compared to traditional methods (e.g., the calculation of the histogram of the inter-signal intervals **[Klump and Gerhardt, 1992]**) **when animal communication shows complicated dynamics.**"

Comment 48:

Line 176. In "paddy field of Kyoto University" do you mean "at Kyoto University" or "belonging to Kyoto University" and located elsewhere?

Response:

Thank you for the comment. The paddy fields are located within the premises of Kyoto University. Therefore, the phrase "at Kyoto University" is more appropriate. We replaced the phrase "of Kyoto University" with "at Kyoto University" (Line 240).

Comment 49:

Lines 178-179. Change "line at the interval" to "line at intervals".

As I indicated above, I'm not sure what you mean by "a stationary room". Most rooms are stationary. Do you mean that the positions of the caged frogs were stationary?

Response:

Thank you very much for the comment. We changed the phrase according to your comment (Line 243). In addition, we deleted the word "in stationary room" from the sentence. As I responded to the 21st comment, we used the word "stationary" to emphasize that condition of the room (e.g., illuminance and temperature) was stationary. However, such a condition is normally stationary in a room. So we consider that it is not necessary to emphasize it here.

Comment 50:

Line 180. Change “by three” to “using three”. You should provide information on the type (manufacturers, models) of recording equipment.

Response:

We changed the phrase according to your comment (Lines 244-246). Here, we added the information on the manufacturers and models for both microphones and recorder. Thank you very much for the comment.

Comment 51:

Line 183-184. Change “44 times in total” to “44 times”. Change “trials of the experiment, some of three” to “trials, some of the three”.

Response:

We changes the phrases (Line 249).

Comment 52:

Line 185. “Change “due to” to “using”.

Response:

We changed the phrase (Line 251).

Comment 53:

Line 186-187. Change “on a sample size of the empirical data, we choose” to “on a sample size of the empirical data, we chose”.

Response:

We modified the verb tense according to your comment (Line 252).

Comment 54:

Some clarification is also needed here. By “all the three frogs” I think you mean that the total number of calls given by the three frogs (calls of frog 1 + calls of frog 2 + calls of frog 3) was required to exceed 1400. Is this the case? Or, was each male required to give 1400 calls?

Response:

Thank you for the comment. Here, we mean that **each male** was required to produce more than 1400 calls. To clarify this point, we replaced the phrase "all the three frogs" with "each male" (**Line 252**).

Comment 55:

Begin a new sentence on line 187 with “The four datasets were”.

Response:

We changed the beginning of the sentence according to your comment (**Line 253**).

Comment 56:

Line 188-190. Change “May in 2009, respectively. Then, the audio data of the four datasets were analyzed by the method of independent component analysis, and were separated into call signals of respective frogs [15]” to “May in 2009. The audio data of each of the four datasets were separated into call signals of individual frogs using independent component analysis [15].”

Response:

Thank you very much for the valuable comment that makes the structure of the sentence much more clear. We changed the sentence according to your comment (**Lines 254 and 255**).

Comment 57:

Line 191. Based on what you state in lines 200-201, I believe that you mean “three chorusing

bouts from 143 chorusing bouts” rather than “three choruses from 143 choruses”. If so, make the necessary change.

Response:

Thank you very much. Yes, the phrase that you suggest appropriately describes what we want to say here and it is much more clear. Therefore, we changed the sentence according to your comment (Line 256).

Comment 58:

Line 192. Delete the word “respective”.

Response:

We deleted the word (Line 258).

Comment 59:

Line 195. Change “guideline” to “guidelines”.

Response:

We changed the word (Line 260).

Comment 60:

Line 199. Change “phase in the range” to “phase to the range”.

Response:

We changed the phrase (Line 271).

Comment 61:

Line 201. Change “calls almost at the interval” to “calls nearly every”.

Response:

We changed the phrase (Line 273).

Comment 62:

Line 203. If this is what you mean, make the following change.

Substitute “incompatible with” for “out of the scope of”.

Response:

Thank you. The phrase that you suggest is more appropriate to describe what we mean here. Then, we changed the phrase according to your comment (Line 275).

Comment 63:

Line 205. Change “and then reset to be 0” to “and then is reset to 0”.

Response:

We changed the phrase (Line 266).

Comment 64:

Line 214. Change “the term of” to “the term for”.

Response:

We changed the phrase (Line 288).

Comment 65:

Line 223. Change “material for detail” to “material for details”.

Response:

We changed the phrase (Line 311).

Comment 66:

Lines 241-242. Change “(1) an inter-frog distance, (2) an inter-call interval, and (3) leader-follower relation-ship. An inter-frog distance” to “(1) the inter-frog distance, (2) the inter-call interval, and (3) the leader-follower relation-ship. The inter-frog distance”

I’m curious. You always used a male-male separation of 50 cm. Have you varied the spacing so that different pairs of males are separated by different distances?

Response:

Thank you for the comment. First, we changed the phrase according to your comment (Lines 362 and 363). Second, as for the male-male separation, we deployed three frogs along a line at intervals of 50 cm between closest neighbors throughout all the experimental trials, meaning that the distance between nearest neighbors was 50 cm while the distance between a distant pair was 100 cm. Therefore, the distance varied between 50 cm and 100 cm depending on pairs of male frogs in our experiment, which allows us to statistically examine the relationship between attention and inter-frog distance utilizing the variance in the distance. To clarify this point, we added a sentence to this paragraph (Lines 364-367). In addition, we added explanation on the male-male separation in the sections of *Results* and also *Materials and Methods* by replacing the phrase "at intervals of 50 cm" with "at intervals of 50 cm between nearest neighbors" (Lines 66 and 243).

Given the limited spatial structure of male frogs provided in this study, however, further studies are required to examine attention among male frogs in other types of spatial arrangement such as non-even distribution in a two-dimensional space as you suggested. Note that we also discuss this point in the revised manuscript (Lines 190-210) as explained in our response to your 37th and 38th comments.

Comment 67:

Line 247. Change “the same chorus” to “the same chorusing bout”.

Response:

We changed the phrase according to your comment (Line 371). In addition, we refer Figure S3 of

supplementary information to help reader to understand the definition of leader-follower relationship assumed in this study (see also our response to your 39th comment).

Comment 68:

Line 250. What you did is a little unclear to me. In any event, I think the following wording is better. If you agree, make the following change.

Substitute “of making each frog the chorus leader” for “for taking a chorus reader for each frog”. Note also that you misspelled “leader” as “reader”.

Response:

Thank you very much. The phrase that you suggest describes more appropriately what we want to say here. We then changed the phrase according to your comment (Line 375). We also appreciate your kind notice on the misspelling. We also corrected it.

Comment 69:

For the next paragraph, the continuous line numbering was absent. So, I will refer to the page-specific line numbers.

Page 13, line 20. Change “areas so as to” to “areas to”.

Response:

We apologize that the paragraph lacked the continuous line numbering as you mention. In the revised manuscript, we have fixed this mistake on the line numbering. Thank you for informing us that.

In addition, we appreciate your comment on the wording. According to your comment, we changed the wording (Line 378).

Comment 70:

Page 13, line 26. Do you mean “and fixed factors” rather than “of fixed factors”? That is, do you mean that the variables you list listed are ‘fixed factors’ in your model?

Response:

Thank you very much for your comment. The previous description was based on our understandings that (1) explanatory variables in a generalized linear mixed model can consist of both fixed factors and random factors, and (2) the variables that we list here (i.e., X_{dis} , X_{int} , and X_{prob}) are fixed factors. Therefore, the phrase "explanatory variables and fixed factors" that you suggest is much more appropriate.

As for this point, however, we further checked relevant studies using GLMM and confirmed that it is more usual to call these variables simply as "fixed factors" or "fixed effects". Therefore, we replaced the phrase "explanatory variables of fixed factors" with "fixed factors" (Line 382), and then added the same modification throughout the manuscript.

Comment 71:

Page 13, line 29. Should this be “variables and random factors” rather than “variables of random factors”?

Change “difficult to be quantified” to “difficult to quantify”.

Response:

Based on our response to your 70th comment, we also replaced the phrase "explanatory variables of random factors" with "random factors" (Line 383). In addition, we changed the phrase to “difficult to quantify” according to your comment (Line 384). Thank you very much for the valuable comments.

Comment 72:

Lines 251-252. You state “To reproduce this feature”. I’m not certain to what feature you are referring.

Response:

Thank you. We agree with your comment pointing out the ambiguity of the phrase. Here, the feature means that the quantified attention (Y_{att}) is restricted to positive value. To clarify this

point, we changed the phrase “To reproduce this feature” to “To reproduce the positive distribution of the response variable Y_{att} ” (Lines 389 and 390).

Comment 73:

Line 257. Change “coefficient is less than” to “coefficient was less than”.

Response:

We modified the verb tense according to your comment (Line 395).

Comment 74:

Line 311. For reference 18, change “begins wells” to “begins Wells”. Wells refers to the last name of Kentwood Wells and so should be capitalized.

Response:

Thank you very much for pointing out it. This was our mistake. We capitalized the last name of Kentwood Wells that is included in the title of the reference (Line 451).

Comment 75:

Line 326. Change “bayesian” to “Bayesian”.

Response:

We capitalized the word according to your comment (Line 465).

Comment 76:

Figure Captions

For Figure 1C, males 1 and 3 do overlap. So you need to rephrase this.

Response:

Thank you for the comment. As for the caption of Figure 1, we added explanation on both clustered anti-phase synchronization and tri-phase synchronization. Accordingly, in the

explanation of clustered anti-phase synchronization, we rephrase that males 1 and 3 overlap their calls.

Comment 77:

Page 18. Line 53 (Caption of Figure 2). Change “dynamics according to” to “dynamics using a”.

Response:

We changed the phrase (see the revised caption of Figure 2).

Comment 78:

Page 20. Line 29. Change “behavioral experiment of actual frogs” to “the behavioral experiment with actual frogs”.

Response:

We changed the phrase according to your comment (see the revised caption of Figure 4).

Comment 79:

Page 22. Lines 33-38. You may need to reword this figure caption based on changes you may need to make that I mention above for Page 13, line 26.

Response:

Thank you. Based on our response to your 70th comment, we changed the phrase to "fixed factors" (see the revised caption of Figure 7).

Comment 80:

References that you should cite which are not already in your references section.

Greenfield, M. D. & Roizen, I. 1993 Katydid synchronous chorusing is an evolutionarily stable

outcome of female choice. *Nature* 364, 618-620.

Greenfield MD, Rand AS (2000) Frogs have rules: Selective attention algorithms regulate chorusing in *Physalaemus pustulosus* (Leptodactylidae). *Ethology* 106:331–347.

Klump, G. M. & Gerhardt, H. C. 1992. Mechanisms and function of call-timing in male-male interactions in frogs. In: *Playback and Studies of Animal Communication* (Ed. by P. K. McGregor), pp. 153-174. New York: Plenum.

Neelon, D. P. and G. Höbel. 2019. Staying ahead of the game—plasticity in chorusing behavior allows males to remain attractive in different social environments.

Behav. Ecol. Sociobiol. 73:124 <https://doi.org/10.1007/s00265-019-2737-1>

Schwartz, J. J. 1993. Male calling behavior, female discrimination and acoustic interference in the Neotropical treefrog *Hyla microcephala* under realistic acoustic conditions. *Behav. Ecol. Sociobiol.* 32: 401-414.

Schwartz, J. J., Buchanan, B. and H. C. Gerhardt. 2002. Acoustic interactions among male gray treefrogs (*Hyla versicolor*) in a chorus setting. *Behav. Ecol. Sociobiol.* 53:9-19.

Vélez A, Schwartz JJ, and Bee MA (2013) Anuran acoustic signal perception in noisy environments. In: Brumm H (ed) *Animal Communication and Noise*. Springer: New York, pp 133-185.

Response:

We appreciate it very much that you provided valuable information on relevant papers. I have carefully read these papers, and agree with your comment that we should refer them. As explained in our responses to your comments, we referred these papers in the revised manuscript as following the context of each paper (The reference numbers are [38-44] in the revised manuscript). Thank you very much again for your valuable comments that greatly helps us to improve the manuscript.

Appendix B

Dear Prof. Wen-Xu Wang and Prof. Kevin Padian,

We are pleased to send you the revised version of our manuscript. We have made all the changes suggested by the reviewers. We feel that the manuscript has improved substantially thanks to the feedback. In addition, we carefully read the whole manuscript and added the following minor updates:

Modification of expressions:

- **Line 160:** We changed "a lower repetition rate" to "a lower repetition rate (i.e., a longer inter-call interval)" in order to clarify the relationship between the repetition rate and inter-call interval.
- **Line 346:** We changed "the distribution of ψ_{nm} " to "the stationary distribution $\hat{f}(\psi_{nm})$ " in order to specify what the distribution is.
- **Caption of Figure S3 in "Suppelementary_Information.pdf":** To simplify the structure of the sentence, we changed "Leader-follower relationship in this study. We assume that the leader, the 1st follower, and the 2nd follower are defined as the frogs that..." to "Leader-follower relationship of chorusing males. In this study, we define the leader, the 1st follower and the 2nd follower as the males that ...".

Correction of typos:

- **Caption of Figure 3:** We changed " $\Gamma_{n,m}$ " to " Γ_{nm} ".
- **Page 2 of "Suppelementary_Information.pdf":** We changed "Data Accessibility" to "Data accessibility".
- **Table S3 of "Suppelementary_Information.pdf":** We changed Equation (11) to Equation (10) in the box of "Stationary distribution" and "Definition and brief explanation". In addition, we changed "the nth frog produce calls" to "the nth frog produced calls" in the box of "Kullback-Leibler divergence" and "Relevance to frog choruses".

A marked-up copy of the changes from the previous manuscript is also attached as a different file (Marked-up-copy.pdf). We greatly appreciate your feedback.

Yours Sincerely,

Ikkyu Aihara (for all the authors)

Response to the comment of Reviewer 1

Comment:

The authors satisfactory responded to my comment. I recommend the manuscript for publication.

Response:

Thank you very much for reading the contents of our revision.

Responses to the comments of Reviewer 2

General comment:

Ota et al. have done an excellent job revising their manuscript according to the comments I made in my review as well as those made by the other reviewer. However, there are a small number of language-related items to which I want to draw the attention of the authors.

Response:

Thank you very much for careful reading of the revised manuscript. As described below, we updated the manuscript according to your comments. We appreciate it very much because all the comments greatly help us to improve English and also the structure of this manuscript.

Specific Comments:

Comment 1:

Line 27: Delete the word “also”.

Response:

Thank you. We deleted the word from the sentence (Line 27).

Comment 2:

Lines 29-20. Change “they are likely to effectively communicate and facilitate assessment of signalers within the aggregation” to
“it may facilitate effective communication and assessment of signalers within the aggregation”.

Response:

We changed the wording according to your comment (Lines 29 and 30).

Comment 3:

Line 35” Change “when forming school” to “when forming a school”.

Response:

We added the indefinite article "a" to the sentence (Line 34).

Comment 4:

Line 106: Change “produce calls at specific interval” to “produce calls at a specific interval”.

Response:

Thank you. We added the indefinite article "a" to the sentence (Line 105).

Comment 5:

Lines 106-107: Change “produce calls at unspecific interval” to “produce calls at an unspecific interval”.

Response:

We added the indefinite article "an" to the sentence (Line 106).

Comment 6:

Line 142: Change the title of this section from “Discussions” to “Discussion”. I failed to notice this error when I first read the manuscript.

Response:

Thank you very much for the comment. We changed the title according to your comment (Line 141).

Comment 7:

Lines 176-178. Change “Given that a chorus leader in our definition can also avoid his calls to be masked by calls of followers at the beginning of a chorusing bout (see Figure S3 in Supplementary Information), we speculate that our result on”

to

“Given that, based on our definition of a chorus leader, a leading male can also avoid masking of his calls by those of followers at the beginning of a chorusing bout (see Figure S3 in Supplementary Information), we speculate that our result on”

Response:

Thank you very much for the valuable comment. We changed the wording according to your comment (Lines 175-178).

Comment 8:

Lines 181-182. Change “experiments using female *H. japonica* is required because acoustic preference of leader-follower relationship in our definition was not examined yet”

to

“experiments using female *H. japonica* are required because an acoustic preference for chorus leaders (our definition) has not yet been examined”.

Response:

We changed the sentence (Lines 180 and 181).

Comment 9:

Line 191: Change “understandings” to “understanding”. Change “we restricted spatial” to “we restricted the spatial”.

Response:

Thank you for the comment. First, we changed the word "understandings" to "understanding" according to the comment (Lines 189 and 190). With respect to the second point, we deleted the phrase because of our response to your 10th comment (the next one). Please read our response described below.

Comment 10:

Lines 190-204. This section is overly wordy and repetitive. There are also some language problems. Please change it to read as follows.

“The application of our methodology to a variety of empirical data would contribute to further understanding of selective attention in frog choruses. In our laboratory experiment, we used a linear arrangement of subjects because male Japanese tree frogs are often positioned along edges of a paddy field [16]. Therefore, this was a reasonable approximation of the actual spatial distribution of the male frogs at a field site. However, unlike the evenly-spaced males in our laboratory experiment, inter-frog distance can vary among linearly arranged males at paddy fields. With respect to this point, even with a non-even distribution of Japanese treefrogs we observed that each pair of nearest neighbors tends to alternate their calls in the field [17].” Nevertheless, because the spatial distribution of male frogs in two and three-dimensions can vary among species and chorusing venues [4], additional investigation of call timing among males under a variety of distribution patterns is certainly warranted. Given that the distance among male frogs profoundly affects the loudness of calls that other frogs recognize, the magnitude of attention is likely affected by the spatial distribution of male frogs. Related to this point...”

Response:

Thank you very much. We carefully read your suggestion and confirm that the suggested paragraph correctly contains what we want to say here and it is much better than the previous one. Therefore, we changed the structure and wording of the paragraph according to your comment (Lines 189-200). We appreciate it very much that your suggestion greatly helps us to improve the paragraph.

Comment 11:

Line 223. Change “On the other hand, technical issue of our methodology needs to be further examined.” to “On the other hand, a technical aspect of our methodology needs to be addressed.”

Response:

We changed the sentence according to your comment (Line 221).

Comment 12:

Line 229. Change “Note that the similar” to “Note that a similar”.

Response:

We changed the article "the" to "a" according to your comment (Line 225).

Comment 13:

Line 232. Change “that noise term” to “that the noise term”. Change “incorporate noisy” to “incorporate the noise”.

Response:

Thank you for the comment. We changed the phrases (Lines 228 and 229).

Comment 14:

Line 290. Change “contain noisy component” to “contain a noise component”.

Response:

We changed the phrase (Line 286).

Comment 15:

Supplementary Material

Table S1. In the box for “Noise” and “Relevance to frog choruses”, change “This term represent” to “This term represents”.

Response:

Thank you very much for your careful reading of the supplementary material. We fixed the mistake according to the comment (see the box for “Noise” and “Relevance to frog choruses” in Table S1).

Comment 16:

Table S2. In the box for “Kullback-Leibler divergence” and “Relevance to frog choruses”, change “at unspecified interval” to “at an unspecified interval”.

Response:

We added the indefinite article "an" to the phrase (see the box “Kullback-Leibler divergence” and “Relevance to frog choruses” for in Table S3). Thank you.

In addition to the above changes, we carefully read the whole manuscript and added the following minor updates:

Modification of expressions:

- **Line 160:** We changed "a lower repetition rate" to "a lower repetition rate (i.e., a longer inter-call interval)" in order to clarify the relationship between the repetition rate and inter-call interval.
- **Line 346:** We changed "the distribution of ψ_{nm} " to "the stationary distribution $\hat{f}(\psi_{nm})$ " in order to specify what the distribution is.
- **Caption of Figure S3 in "Suppelementary_Information.pdf":** To simplify the structure of the sentence, we changed "Leader-follower relationship in this study. We assume that the leader, the 1st follower, and the 2nd follower are defined as the frogs that..." to "Leader-follower relationship of chorusing males. In this study, we define the leader, the 1st follower and the 2nd follower as the males that ...".

Correction of typos:

- **Caption of Figure 3:** We changed " $\Gamma_{n,m}$ " to " Γ_{nm} ".
- **Page 2 of "Suppelementary_Information.pdf":** We changed "Data Accessibility" to "Data accessibility".
- **Table S3 of "Suppelementary_Information.pdf":** We changed Equation (11) to Equation (10) in the box of "Stationary distribution" and "Definition and brief explanation". In addition, we changed "the nth frog produce calls" to "the nth frog produced calls" in the box of "Kullback-Leibler divergence" and "Relevance to frog choruses".

A marked-up copy of the changes from the previous manuscript is also attached as a different file (Marked-up-copy.pdf). We greatly appreciate your feedback.